# Selective suppression of antisense transcription by Set2-mediated H3K36 methylation

Swaminathan Venkatesh[1], Hua Li[1], Madelaine M. Gogol[1] & Jerry L. Workman[1]

Maintenance of a regular chromatin structure over the coding regions of genes occurs co-transcriptionally via the 'chromatin resetting' pathway. One of the central players in this pathway is the histone methyltransferase Set2. Here we show that the loss of Set2 in yeast, *Saccharomyces cerevisiae,* results in transcription initiation of antisense RNAs embedded within body of protein-coding genes. These RNAs are distinct from the previously identified non-coding RNAs and cover 11% of the yeast genome. These RNA species have been named Set2-repressed antisense transcripts (SRATs) since the co-transcriptional addition of the H3K36 methyl mark by Set2 over their start sites results in their suppression. Interestingly, loss of chromatin resetting factor Set2 or the subsequent production of SRATs does not affect the abundance of the sense transcripts. This difference in transcriptional outcomes of overlapping transcripts due to a strand-independent addition of H3K36 methylation is a key regulatory feature of interleaved transcriptomes.

[1] Stowers Institute for Medical Research, 1000 E. 50th Street, Kansas City, Missouri 64110, USA. Correspondence and requests for materials should be addressed to J.L.W. (email: jlw@stowers.org).

Advances in genomics technologies have led to the system-wide identification of transcription units in several organisms. One feature common to transcript profiles from different organisms is its pervasiveness—a consequence of transcription occurring from large parts of the genome in addition to the protein-coding genes[1]. About 80% of the human genome is transcribed, while <2% is coded into proteins[2]. Single-celled organisms like yeast transcribe about 85% of their genome with 22% constituting protein-coding transcripts[3]. Despite their widespread production, transcript abundances vary widely as they are targets of RNA-degradation machineries. The abundance of a transcript is determined by the balance between its rate of production and degradation. Identification of novel transcripts with low abundance has remained a challenge, leading to the conclusion that current estimates about the extent of pervasive transcription are conservative in nature.

To offset the limited improvements in the sensitivity to which low abundant RNA can be confidently identified, current strategies are designed to enhance RNA stability, so that they fall within the detection range of the latest next-generation sequencers. One method is to disrupt the RNA-degradation pathways, leading to increased abundance of these transcripts, aiding subsequent sequencing using the highly sensitive microarray or Next-generation sequencing techniques[4–7]. This approach has resulted in the discovery of numerous transcripts, predominantly those that arise from divergent promoters and over intergenic regions.

Along with degradation, a number of other mechanisms work to limit pervasive transcription. A recent screen[8] identified a number of factors that alter chromatin and regulate the production of non-coding RNA (ncRNA) implicating chromatin dynamics, particularly histone turnover at promoters, in controlling divergent transcription initiation. Histone modifications, especially acetylation of key residues, facilitated this process along with chromatin remodelers like the Swi/Snf complex. On the other hand, the Isw2 chromatin remodelling complex was shown to limit the production of antisense transcripts[9] by preventing chromatin dynamics over promoters.

The process of chromatin dynamics over promoters is well understood, with the identification of key factors and elucidation of molecular mechanisms[10]. Recent work has described the factors and mechanisms necessary to limit disruption and support re-formation of chromatin during transcription elongation. One such pathway involves the histone methyltransferase Set2-mediated H3K36 methylation. Association of Set2 with the phosphorylated C-terminal tail of RNA Polymerase II (RNAPII)[11], recruits the enzyme over coding regions during transcription elongation and facilitates trimethylation of histone H3K36 (ref. 12). This methyl mark maintains the integrity of the nucleosome by preventing histone exchange[13], thereby limiting chromatin dynamics. It also recruits the Isw1b chromatin remodelling complex to ensure proper spacing of nucleosomes[14], and the Rpd3S deacetylase complex to remove acetyl marks on nucleosomes over the gene bodies[15–17]. Studies have also shown that Isw1-catalysed spacing of chromatin is necessary for the Rpd3S-mediated deacetylation of neighbouring nucleosomes[18]. These processes together constitute the chromatin-resetting pathway[10], that reorganizes chromatin after transcription elongation. Disruption of the resetting mechanism leads to the production of internally initiated transcripts for selected genes[17,19]. This chromatin-based regulation of ncRNA is a common feature in yeast[20,21] and is believed to be conserved in humans.

Antisense transcripts are a class of ncRNA that are transcribed from the strand complementary to protein-coding genes or non-coding transcription units (often called the sense strand)[22].

Interestingly, antisense transcription accounts for over 30% of the human transcriptome[23]. Although the yeast genome is known to transcribe the antisense strand based on native elongating transcript sequencing (NET-Seq) data[24], the identity of these transcripts are unknown as these RNAs are very unstable. The exact mechanism regulating the transcription initiation of these antisense transcripts is also as yet unclear.

Using deep sequencing methods, we have identified and annotated a novel group of antisense transcripts, called Set2-repressed antisense transcripts (SRATs), which are de-repressed upon loss of Set2. These transcription units are usually buried within the coding region of genes. We show that elongation over the promoters of overlapping transcription units prevents chromatin dynamics, suppressing transcription initiation from these promoters. Despite an increased expression of these internal antisense units, sense transcription elongation is largely unaffected in the SET2 deletion mutant (set2), indicating strand-specific effects of strand-independent addition of histone modifications. Identification of these transcription units would serve as a valuable resource to understand both the function and regulation of ncRNA.

## Results

**Identification and validation of SRATs**. To identify the transcripts produced upon the deletion of SET2, we purified total RNA from the deletion and wild-type yeast strains in replica. With the goal of discerning RNA production from both strands of DNA, we prepared libraries from ribodepleted total RNA using the strand-specific RNA sequencing kits (Illumina), and subjected the libraries to deep sequencing (Methods). High level of correlation between the replicates attested the reproducibility of the samples.

On the basis of a bioinformatics identification pipeline (Methods), we identified a list of 853 antisense transcripts that were de-repressed upon loss of the Set2 histone methyl-transferase. We named these SRATs. SRAT282 is an example, found associated with YDR452W (Fig. 1a). We compared the abundance of these selected SRATs in the wild-type and deletion mutants, and found significant upregulation ($P$ value$<2.2e-16$, Welch two-sample test) of SRAT expression upon SET2 deletion (Fig. 1b, left).

One common strategy employed by the cell to protect the RNA from degradation and thereby enhance their stability, is the addition of an oligo (dA) tail at the 3′ end of a nascent transcript. To measure of the stability of the SRATs, we determined whether some of these transcripts were polyadenylated (Methods). The poly-A-enriched fraction was subjected to strand-specific RNA-seq, with a similar genomic coverage as for the ribodepleted total RNA samples. Interestingly, 532 SRATs are polyadenylated, accounting for almost 50% of the SRATs identified (Supplementary Data 1). These selected polyadenylated SRATs show an increased abundance upon SET2 deletion (Fig. 1b, right), including SRAT282 (Fig. 1a).

Some SRATs identified by bioinformatics analysis were validated by strand-specific northern blots (Methods), with radiolabelled riboprobes against four specific SRATs generated from our annotation. All four SRATs demonstrated distinct Set2-dependent expression (Fig. 1c, right) both in total RNA and poly-A-enriched samples. Although SRAT282 shows enhanced expression in a set2 mutant, it was detected at very low levels in the wild-type samples as well (Fig. 1c, left; Supplementary Fig. 1). Furthermore, we found that SRAT086, which is found at a much lower abundance compared with the other SRATs, also possesses the poly-A tail (Fig. 1c, right). These results indicate that variations in the relative abundance of SRATs are primarily at the level of transcription initiation and not stability.

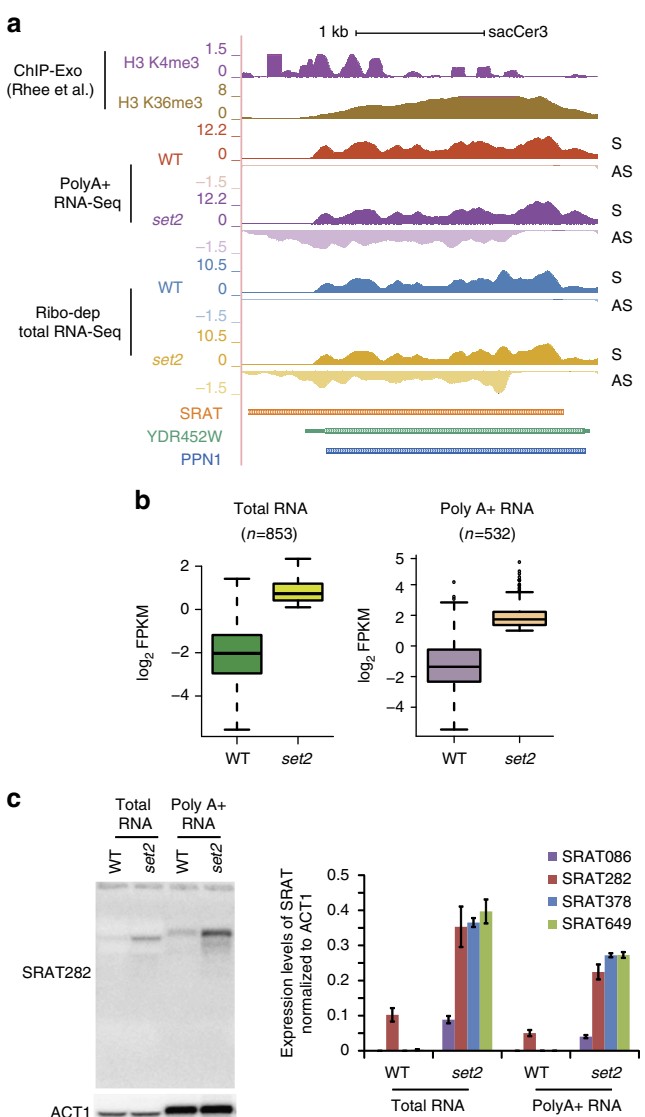

**Figure 1 | Identification of Set2-repressed antisense transcripts (SRATs).** (**a**) Genome browser profile showing the distribution of modifications and the transcripts (both ribodepleted and Poly A+ RNA) produced in wild-type strain (WT) and the SET2 deletion mutant (*set2*) over the gene *YDR452W* (in green including the untranslated regions and the standard name, PPN1 shown in blue denoting the protein-coding region). The modifications are reanalysed tracks from[33]. One replicate for each ChIP-Exo track of H3K4me3 and H3K36me3 distributions are shown here. Each track, an amalgamation of two biological repeats, is separated into the sense strand (S) on top, running from left to right and the antisense strand (AS) in the bottom running from right to left. The SET2 deletion mutant produces enhanced levels of SRAT282 from within the gene body. (**b**) Boxplot showing the abundance of the SRATs (log2 FPKM) in the WT and *SET2* deletion mutants from total RNA (left) and poly-A-enriched RNA (right). The total number of SRATs used in the analysis is denoted above each plot. (**c**). (Left) Strand-specific northern blot probing for SRAT282 using either total RNA or polyadenylated RNA in wild-type (WT) and *SET2* deletion mutant (*set2*). ACT1 is used as a loading control. The aberrant mobility of the SRAT between the total RNA and polyA-enriched samples is due to large amounts of rRNAs in the total RNA samples. (Right) Quantitation of strand-specific northern blots indicating the expression level of selected SRATs, normalized to the level of *ACT1* in total RNA and polyadenylated mRNA. Error bars denote the s.e.m. of three independent repeats.

**Expression of Set2 in a *set2* strain represses SRATs.** To confirm that the identified SRATs were indeed the consequence of the deletion of *SET2*, we transformed both wild-type and the *set2* mutant strains with either an empty vector or a full-length Set2 yeast expression clone[25]. We purified total RNA from these strains and carried out the strand-specific RNA-Seq as described previously. We confirmed the loss of Set2 expression in the deletion strains by RNA-Seq (Supplementary Fig. 2A,B), and by following the H3K36 methylation levels in the strains by immunoblotting (Supplementary Fig. 2C). As expected, we found that the expression of full-length Set2 in the strain deleted for the same, resulted in repression of the SRAT282 (Supplementary Fig. 3A). The wild-type strains were unaffected by the overexpression of Set2 (Supplementary Fig. 2C). The SRAT density profiles show that the expression of a functional Set2 methyltransferase in the *set2* strain resembles the wild-type strain (Supplementary Fig. 3B). The density traces compare the various replicates with each other denoting a good parity between the duplicates.

We selected 752 SRATs that were most significantly altered in the *set2* mutant with the empty vector (similar to the *set2*) compared with the deletion strain with the full-length expression vector (similar to the wild-type; Supplementary Fig. 3C). We compared this to our previously identified list of 853 SRATs and found an 86% overlap (Supplementary Fig. 3D). The lack of a complete overlap may be in part due to variations in the two independent experiments and the strict statistical cutoffs we instituted at each stage.

This reproduction of the data suggests that identical SRATs are produced in different strains—initiating from specific internal promoter sites and are not the result of random initiation events over a broad locus. Finally, we carried out strand-specific northern blot assays with probes targeting either SRAT 282 or SRAT 378, on total RNA and validated that Set2 expression in a *set2* strain suppresses the SRAT production (Supplementary Fig. 3E).

**Set2-mediated H3 K36 methylation suppresses SRATs.** The finding that SRAT production switches off when a functional copy of Set2 is introduced into the deletion strain, implicates a central role for this protein in suppressing these transcripts. To understand the mechanism of Set2-mediated suppression of antisense transcription, we focused on its methyltransferase activity. The only known substrate for the methyltransferase activity of Set2 in yeast is the H3K36 residue[26]. Set2 binds to elongating RNA polymerase II (RNAPII) and co-transcriptionally methylates this residue[12]. H3K36 methylation is enriched over the mid to 3′ end of coding regions due to the association of Set2 with the serine-2 phosphorylated form of RNAPII (refs 11,27).

To determine whether H3K36 methylation plays a role in suppressing SRATs, we probed the H3K36A point mutant (that lacks H3 K36 methylation, Supplementary Fig. 4A) using strand-specific RNA-Seq experiments. The results were compared with those obtained from the histone shuffle strain with a wild-type copy of histones on the plasmid (YBL). The H3K36A point mutant also generates SRATs (Fig. 2a), suggesting that H3K36 methylation plays a critical role in suppressing SRAT production. Applying the same cutoffs as described earlier, we selected 893 SRATs that were significantly upregulated in the mutant strain (Fig. 2b). We validated the results by strand-specific northern blots (Fig. 2c), underlining the reproducible production of the SRATs in the different mutants tested. 79% of the SRATs identified in the H3K36A strain overlapped with those previously identified in *set2* (Fig. 2d). On the basis of this data, we can clearly conclude that Set2-mediated H3K36 methylation is responsible for suppressing antisense transcripts.

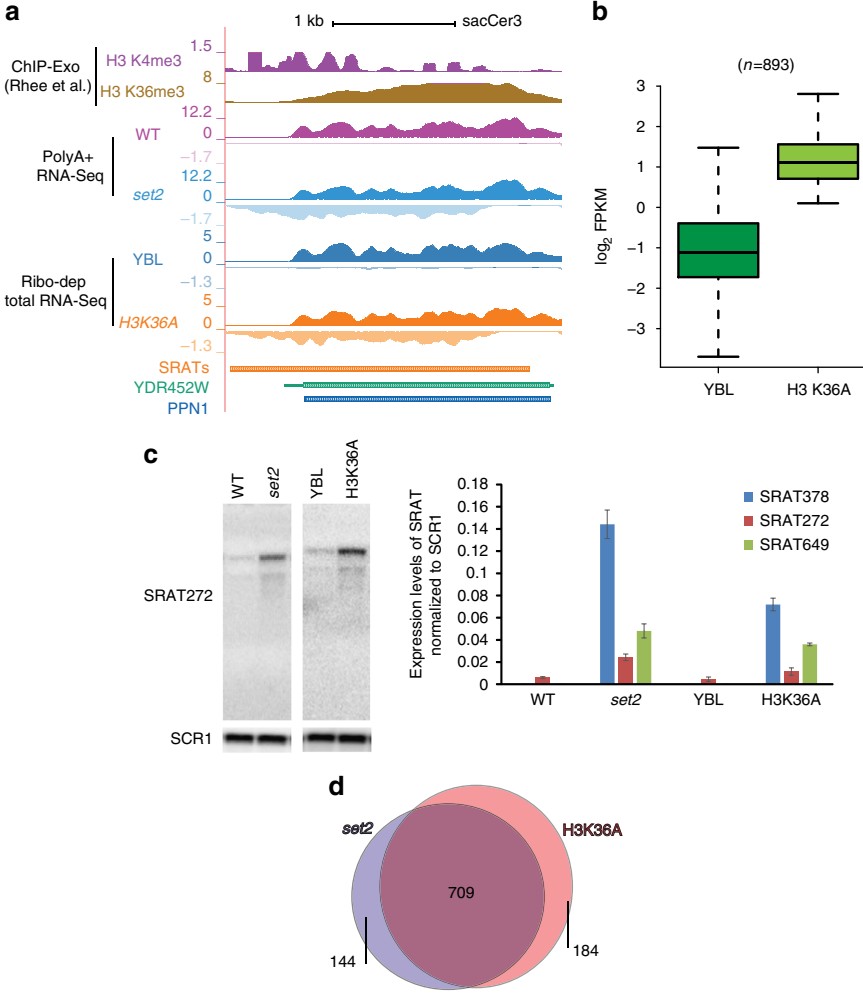

**Figure 2 | Loss of H3K36 methylation de-represses SRAT production.** (**a**) Genome browser profile showing the histone modifications and transcripts (poly A + and ribodepleted) produced in wild-type strain (WT, YBL) and the *SET2* (*set2*) and H3K36A mutants (H3K36A) over the gene *YDR452W* (PPN1). Each track, an amalgamation of two biological repeats, is separated into the sense strand (S) on top, running from left to right and the antisense strand (AS) in the bottom running from right to left. The H3K36A mutant produces the antisense transcript, SRAT282, as observed in the *set2* mutant. (**b**) Boxplot showing the abundance of significant SRATs (log2 FPKM) in the WT (YBL) and H3K36A mutant. The total number of SRATs used in the analysis is denoted above the plot. (**c**) (Left) Strand-specific northern blot probing for SRAT282 using total RNA in either the respective wild-type (WT-BY4741 or YBL), *SET2* deletion mutant (*set2*) and the H3K36A mutant strain. *SCR1* is used as a loading control. (Right) Quantitation of strand-specific northern blots indicating the expression level of selected SRATs, normalized to the level of *SCR1* in total RNA in the indicated mutants. Error bars denote the s.e.m. of three to four independent repeats. (**d**) Venn diagram showing the overlap of statistically significant SRATs produced upon deletion of *SET2* (*set2*) with those in an H3K36A point mutant.

Interestingly, we found a twofold increase in the abundance of SRATs in the wild-type histone shuffle strain (YBL) when compared to the BY4741 wild-type *S cerevisiae* strain (Supplementary Fig. 4B). This increase in abundance of transcripts in the wild-type histone shuffle strain leads to more muted fold-change values compared with the BY4741 strain (Supplementary Fig. 4C). Comparing the fold-change in the *SET2* mutant with that of the H3K36A mutant for each individual SRAT in a scatter plot revealed this bias more clearly (Supplementary Fig. 4D). However, plotting the abundance of each SRAT between the two mutant strains show a comparable increase in both cases (Pearson's coefficient of correlation = 0.75). We attribute this leaky expression of the antisense transcripts in the wild-type shuffle strain to the decreased copy number of the histone genes (one versus the usual two copies found in yeast). A recent publication also showed that decreasing the number of histone copies in yeast leads to a slight alteration in nucleosome distribution[28]. However, the H3K36A mutant recapitulates the

abundance of SRATs to a level similar to the *SET2* deletion mutant, suggesting that the initiation of transcription in these mutants occurs similarly.

**SRATs are buried within coding regions**. We wanted to compare the genomic spread and transcript lengths of the SRATs with previously identified coding and non-coding transcripts (Table 1). Protein-coding genes and the associated untranslated regions (UTRs) cover the 82% of the yeast genome[29]. The cryptic unstable transcripts (CUTs), associated with divergent promoters, occupy 5% of the genome and have a median length of 148 nucleotides. Multiple mechanisms limit their transcription[4,5]. One such mechanism is the Nrd1-Nab3-Sen1 pathway[30]. Nuclear depletion of the termination factor, Nrd1 leads to production of Nrd1-unterminated transcripts (NUTs)[7]. These transcripts are a result of unterminated transcription of CUTs, stable unannotated transcripts (SUTs) and the Xrn1-sensitive unstable

transcripts (XUTs), accounting for both the observed genomic coverage (22%) and transcript length (1,528 nucleotides). The XUTs, which are a mixture of both intergenic and antisense transcripts[6], cover 12% of the genome with a median length of 640 nucleotides. Overall, the SRATs occupy 11% of the yeast genome with a median length of 883 nucleotides (Table 1).

In contrast to the unstable non-coding transcripts, which can only be identified by enhancing RNA stability or production, SUTs[4] constitute a set of stable transcripts that are produced in the wild-type strain. These transcripts cover 6% of the genome and have a median length of 816 nucleotides.

On the basis of the per cent of the genome covered by each class of transcript, we can conclude that most of these transcripts overlap with each other, giving rise to an interleaved transcriptome. Given that the SRATs are predominantly antisense to previously identified transcripts, we wanted to decipher their organization with respect to these transcription units. For this purpose, we focused on three parameters as shown in Fig. 3a. The first parameter is the length of genes that overlap with the SRATs (Gene length). Second, the distance of the antisense start site from the transcription start site (TSS) of the overlapping gene (distance to TSS) and third, the length of the antisense transcript itself. We find that the SRATs are associated with longer genes—with a median transcript length of 2,200 bp. On the basis of the distance to TSS measure, which is less than the length of the gene, we can conclude that the SRATs predominantly initiate within the coding regions of these genes, with a bias towards the 3′ end. Interestingly, the antisense transcript length is usually shorter than the distance to the TSS measure, suggesting that SRATs usually terminate within the gene bodies. We have noticed a few cases where initiation occurs from the 3′-UTR of genes, and some

**Table 1 | Tabular representation of the genome coverage and median transcript lengths of the different classes of transcripts produced in *S. cerevisiae*.**

| Transcript type | % of yeast genome transcribed | Transcript length (nt) |
|---|---|---|
| Coding genes (+UTR)* | 82 | 1,254 |
| CUTs | 5 | 148 |
| SUTs | 6 | 816 |
| XUTs | 12 | 640 |
| NUTs | 22 | 1,528 |
| SRATs | 11 | 883 |

CUTs, cryptic unstable transcripts; SRATs, Set2-repressed antisense transcripts; SUTs, stable unannotated transcripts; UTR, untranslated region; XUTs, Xrn1-sensitive unstable transcripts; NUTs, Nrd1-unterminated transcripts.
*Note that the genome coverage and transcript length for the protein-coding genes includes the 5′ and 3′-UTR.

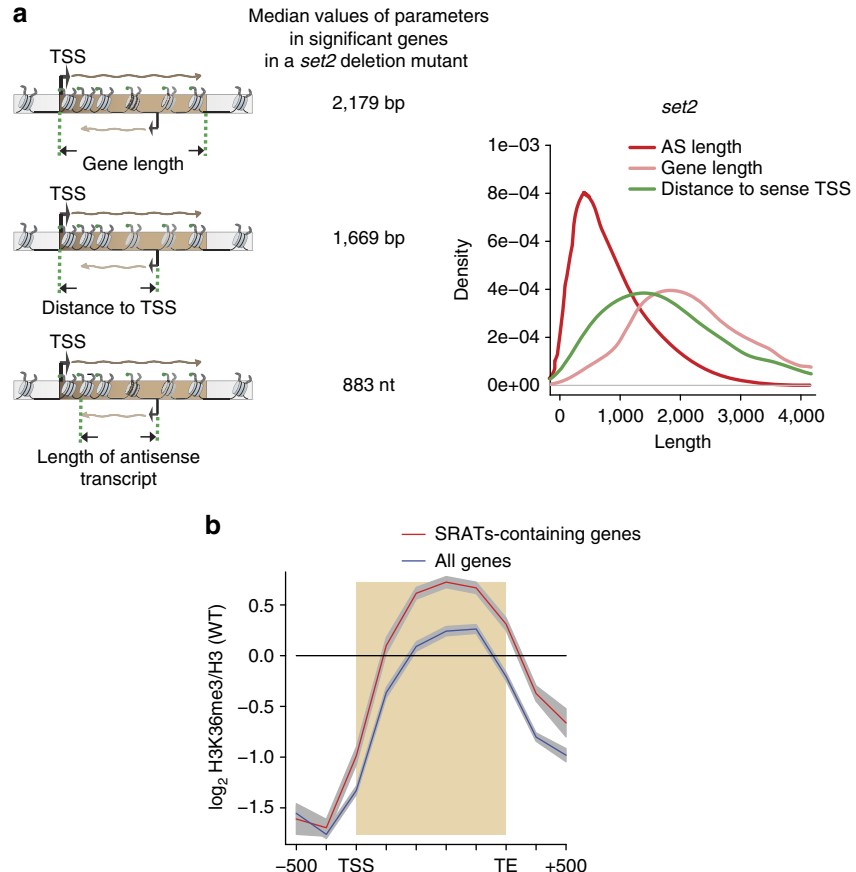

**Figure 3 | SRATs initiate and terminate within the coding regions of genes.** (**a**) On the left, a schematic representation of the three parameters used to define the SRAT relative to the coding region of genes—Gene length, distance to transcription start site (TSS) and the antisense length, and the median values based on our analysis from two independent repeats (see Methods). On the right, the density plots of the three parameters for the SRATs and their associated genes in the *SET2* deletion mutant. (**b**). Gene average plot of the distribution of H3K36me3 normalized to H3 occupancy levels over a metagene from three independent biological repeats. The blue line traces the distribution over all coding genes in yeast, while the red line traces the distribution of H3K36me3 over the protein-coding genes that produce SRATs in a *SET2* deletion mutant (*n* = 853). The grey shading around each line indicates the 95% confidence interval.

cases where the SRAT transcribes over the promoter of the overlapping gene. In addition, longer genes (>5 kb) possess multiple antisense transcripts. A majority of SRATs have transcript lengths under 1,500 nucleotides (Fig. 3a, right), well within the median gene length of 2,200 bp. On the basis of these observations we can conclude that most SRATs generated upon the loss of *SET2*-mediated H3K36 methylation, initiate from the 3′ end of genes and terminate within the coding region.

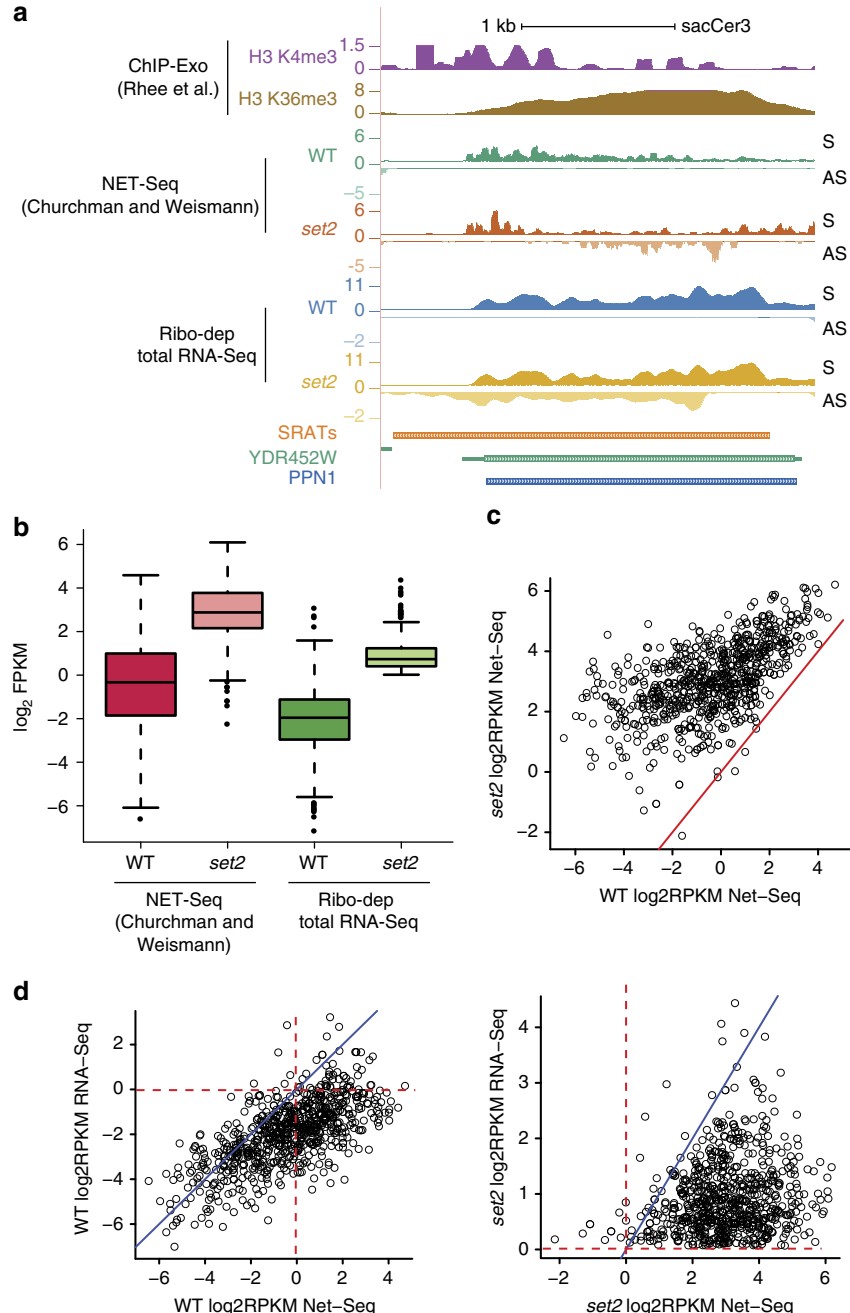

**Figure 4 | SRATs are actively transcribed by RNA Pol II in the wild-type strain and enhanced in the *SET2* deletion strain. (a)** Genome browser profile showing the histone modifications and transcripts (NET-Seq and ribodepleted) produced in wild-type strain (WT) and the *SET2* (*set2*) over the gene *YDR452W* (PPN1). Each track is separated into the sense strand (S) on top, running from left to right and the antisense strand (AS) in the bottom running from right to left. The *set2* mutant produces the antisense transcript, SRAT282, which is also seen in the NET-Seq lanes. (**b**). Boxplot comparing the abundance of a common set of 763 SRATs (log2 FPKM) in the WT and Set2 deletion (*set2*) mutants for the NET-Seq[24] and our ribodepleted RNA-Seq datasets (an amalgamation of two biological repeats). (**c**) Scatter plots denoting the NET-Seq transcript abundance of 763 SRATs in the WT versus the *set2* mutant. The respective values from the *SET2* deletion mutant are distributed on the y axis, while those from the WT are distributed on the x axis. The red line denotes the values where x = y. (**d**). Scatter plots comparing the NET-Seq transcript abundance with that of RNA-seq of 763 SRATs in the WT (left) and *set2* mutant (right) strains. The respective values from the RNA seq are distributed on the y axis, while those from the NET-Seq data are distributed on the x axis. The blue line denotes the values where x = y. The red horizontal and vertical lines denote x or y = 0, which corresponds to an RPKM of 1. Points lying to the right of the vertical lines denote enrichment in the NET-Seq sample, while points lying above the horizontal red line denote enrichment in the RNA-Seq samples.

Given a critical role of H3K36 methylation in preventing the initiation of antisense transcription, we wanted to determine whether the levels of this methyl mark was altered in the genes with buried antisense transcripts compared with all the genes in the yeast genome. For this purpose, we analysed the ChIP-microarray data of the distribution of H3K36 trimethylation normalized to the H3 levels that we published previously[13]. Indeed, we find that genes with antisense transcripts have higher levels of H3K36 trimethylation over the coding regions of the genes (Fig. 3b). This suggests that one of the core functions of the H3K36 methylation mark over gene bodies, under the conditions we tested, is the suppression of spurious transcription initiation.

To determine whether the SRATs are actively transcribed upon loss of Set2, we analysed the previously published NET-Seq data[24]. We find a significant NET-Seq signal associated with SRAT282 in the *set2* mutant (Fig. 4a). We then selected 763 SRATs with significant signal intensity in the NET-Seq experiment and found an increase in their NET-Seq reads upon loss of Set2 (Fig. 4b,c). This clearly shows that the SRATs are transcribed by RNAPII, with an increased engagement in the *set2* mutant versus the wild-type. Approximately half of the SRAT promoters are found to initiate from the nucleosome-free regions (NFR) at the 3′ end of genes, while the other half initiates from within the body of genes (Supplementary Fig. 5). While the

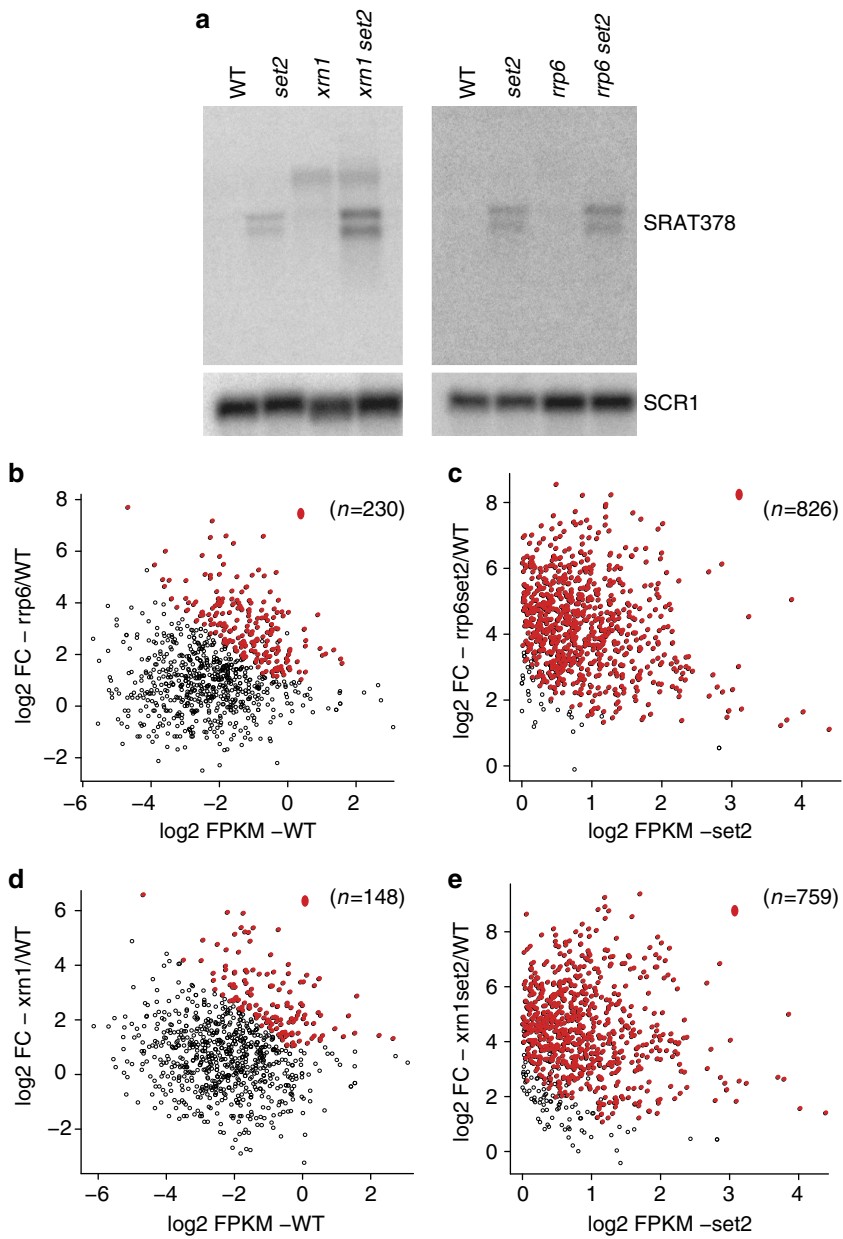

**Figure 5 | SRAT stability enhanced upon loss of RNA-degradation machinery. (a)** (Left) Strand-specific northern blot probing for SRAT378 using total RNA in either the wild-type (WT-BY4741), *SET2* deletion mutant (*set2*), *RRP6* deletion mutant (*rrp6*) and the *RRP6 SET2* double deletion mutant (*rrp6set2*). (Right) Strand-specific northern blot probing for SRAT378 using total RNA in either the wild-type (WT-BY4741), *SET2* deletion mutant (*set2*), *XRN1* deletion mutant (*xrn1*) and the *XRN1 SET2* double deletion mutant (*xrn1set2*). *SCR1* is used as a loading control. (**b**–**e**). Scatter plots comparing the strand-specific RNA-Seq transcript abundance of SRAT a strain compared with the fold change in SRAT expression in the indicated mutant strains. The fold change of RNA abundance in indicated mutants with respect to the wild-type are distributed on the *y* axis, while RNA abundance of the wild-type (WT) (**b**,**d**) or *SET2* deletion (*set2*) (**c**,**e**) are distributed on the *x* axis. The red dots indicate the SRATs that are significantly upregulated in the mutant on the *y* axis. The total number of SRATs used in the analysis is denoted above each plot.

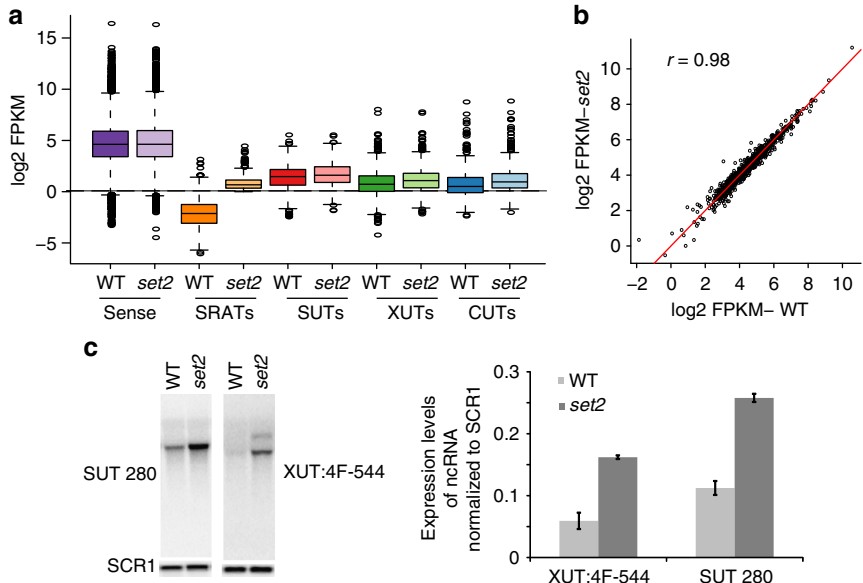

**Figure 6 | Effect of *SET2* deletion on ncRNA production in yeast.** (**a**). Boxplots showing the abundance of the different RNA species as indicated, produced in either the wild-type or *SET2* deletion yeast strain. (**b**) Scatter plot with the abundance of sense protein-coding genes (Log2 FPKM) of wild-type in the *x* axis and *SET2* deletion mutant in the *y* axis. The red line denotes the values where $x = y$. The Spearman coefficient of correlation is provided. (**c**) (Left) Strand-specific northern blot probing for SUT280 and the XUT, 4F-544 using total RNA in either the respective wild-type or *SET2* deletion mutant (*set2*). *SCR1* is used as a loading control. (Right) Quantitation of strand-specific northern blots indicating the expression level of selected ncRNAs, normalized to the level of *SCR1* in total RNA under the indicated conditions. Error bars denote the s.e.m. of three independent repeats.

**Table 2 | Tabular representation of the subset of CUTs, XUTs, SUTs and NUTs that show enhanced abundance in a *SET2* deletion mutant.**

| ncRNA subtype | No. of ncRNA affect by loss of Set2 | % of ncRNA* |
|---|---|---|
| CUTs | 52 | 5.7 |
| SUTs | 48 | 5.9 |
| XUTs | 121 | 7.2 |
| NUTs | 126 | 8.2 |

CUTs, cryptic unstable transcripts; SUTs, stable unannotated transcripts; UTR, untranslated region; XUTs, Xrn1-sensitive unstable transcripts; NUTs, Nrd1-unterminated transcripts.
*The number of ncRNAs showing an increase in expression upon *SET2* deletion is expressed as a per cent of the total number of ncRNA present in each class.

SRATs initiating from the 3′ ends have a well-formed NFR in both wild-type and the *set2* mutant, those within the gene have no discernable NFR. Despite the presence or absence of the NFR, the loss of *set2* results in an accumulation of the active H3K4 trimethyl mark in both the categories, and increased SRAT transcription (Supplementary Fig. 5D,E). To address whether the SRAT promoters are bi-directional in nature, we selected chromosomal regions from the SRAT start site to 500 bp upstream of it, on the sense strand. Quantifying the reads within this window, we found that there was a small but significant increase in the FPKM values between wild-type and *set2* (Welch's two-sample *t*-test, P value = 4.762e − 15; Supplementary Fig. 5A). A scatter plot between the NET-Seq signals in the wild-type versus the fold-change in the *set2* mutant reveals that 310 SRATs have a greater than twofold increase in the upstream reads (Supplementary Fig. 5B), suggesting that these promoters could be bi-directional in nature. Twenty-four of these upregulated regions had no signal associated with them in the wild-type stain, implying that these promoters were definitely bi-directional in nature.

Interestingly, the NET-Seq signals for the SRATs are enriched compared with the ribodepleted RNA-Seq signals in the wild-type strain (Fig. 4d), suggesting that while SRATs are transcribed in the wild-type strain, they are rapidly degraded by various RNA-degradation machineries. This mechanism seems to be active even in the *set2* mutant as the NET-Seq transcripts are more enriched than the RNA-Seq abundances (Fig. 4d, right). This observation indicates that the RNA production and degradation machineries for SRATs are independent of one another.

To confirm whether the SRATs are actively transcribed in the wild-type strain, we targeted RNA-degradation machineries in the cell and carried out strand-specific RNA-Seq. Disruption of RNA-degradation machineries results in increased abundance of a few SRATs (Fig. 5a,b,d, Supplementary Fig. 6). Deletion of either *RRP6* or *XRN1* in combination with *SET2* results in a global increase in the abundance of SRATs (Fig. 5c,e). These data suggest that the SRATs are actively degraded.

**Set2 suppresses antisense transcription initiation.** A comparison of the abundance distribution of each class of transcripts produced either in the wild-type or *set2* mutant reveals some interesting features (Fig. 6a). First, SRATs are the only class of transcripts that show a striking upregulation in their expression upon loss of Set2. The other classes of transcripts are affected to a more limited extent. Second, protein-coding transcripts are about 15–30-fold more abundant than any other class of transcripts[2]. The non-coding transcripts are targets of RNA-degradation pathways, which serve to limit their abundance[4–7].

One feature that stood out in the distribution of sense protein-coding transcript abundance was the lack of change between the *set2* mutant and the wild-type (Fig. 6a). We selected sense genes having polyadenylated transcript abundance above a cutoff (FPKM > 5) in the strains, a false discovery rate (FDR) < 5% and a twofold change (either up- or downregulated) in expression. On the basis of these selection parameters we found 92 upregulated genes and 45 downregulated genes (Supplementary Fig. 7A). We went on to ask whether genes with SRATs buried within their coding regions, show a change in expression upon SRAT de-repression. To our surprise, we found

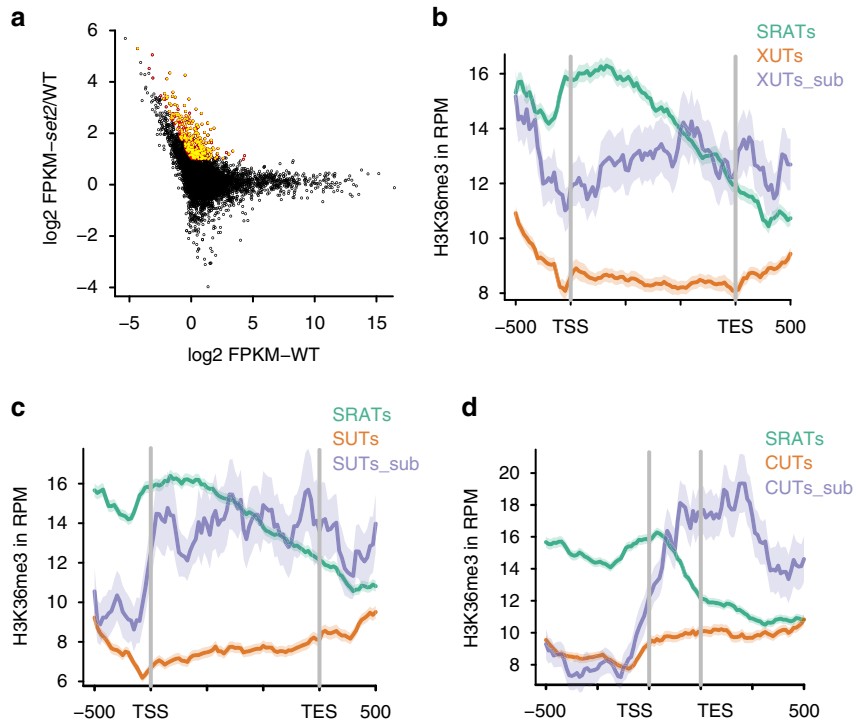

**Figure 7 | Expression of overlapping sense genes suppresses antisense transcription by H3K36 methylation.** (**a**) Scatter plot denoting the wild-type transcript abundance of CUTs, SUTs and XUTs on the *x* axis, and the fold change of each of these transcripts in the *SET2* deletion mutant on the *y* axis. Transcripts that are significantly upregulated using the defined cutoffs are marked as red circles ($n = 347$). Transcripts that have an overlapping protein-coding gene overlapping the ncRNA are marked with filled yellow circles. (**b**). Metagene plots denoting the distribution of H3K36 me3 mark over the promoters and gene bodies of SRATs (green), XUTs (orange), and a subset of XUTs (XUT_sub, $n = 121$) that are upregulated upon loss of Set2 (blue). (**c**) Metagene plots denoting the distribution of H3K36 me3 mark over the promoters and gene bodies of SRATs (green), SUTs (orange) and a subset of SUTs (SUT_sub, $n = 48$) that are upregulated upon loss of Set2 (blue). (**d**) Metagene plots denoting the distribution of H3K36 me3 mark over the promoters and gene bodies of SRATs (green), CUTs (orange) and a subset of CUTs (CUT_sub, $n = 52$) that are upregulated upon loss of Set2 (blue). The lighter areas surrounding the traces in all three figures denote the 95% confidence interval of the traces.

that the 853 genes identified with SRATs, showed no change in sense gene expression in the *set2* mutant (Fig. 6b). The high degree of correlation between the abundance in the wild-type versus the mutant (Pearson's coefficient of correlation = 0.98) suggests that the expression of SRATs in the antisense direction does not affect the sense transcription. These data indicate that under the YPD growth conditions, Set2-mediated H3K36 methylation selectively suppresses antisense promoters.

We next tested whether the other classes of antisense transcripts were affected by the loss of Set2. Interestingly, about 5–10% of all non-coding transcripts identified to date are affected by the loss of H3K36 methylation (Table 2). We validated the change in expression for selected SUTs and XUTs using strand-specific northern blot and quantified the changes (Fig. 6c). We found that SUT 280 was expressed in the wild-type strain, with a 2.5-fold increase in expression upon loss of Set2. The XUT, 4F-544 is expressed in the *set2* mutant alone. These data confirm that the expression of a subset of ncRNA is suppressed by Set2-mediated H3K36 methylation.

**Co-transcriptional suppression of ncRNA by H3K36 methylation.** To understand how Set2-mediated H3K36 methylation affects the expression of selected ncRNA, the abundance of these ncRNA (SUTs, CUTs and XUTs) in the wild-type strain were plotted against the fold-change in ncRNA expression in the *SET2* deletion mutant versus the wild-type (Fig. 7a). We then selected the genes with significant change in expression (Table 2; marked as red). Interestingly, ncRNAs that were poorly expressed (log FPKM value <1) in the strain with a functional copy of Set2,

were most significantly upregulated in the deletion strain (Fig. 7a). This suggested that H3K36 methylation was suppressing the expression of these transcripts in the wild-type strain. Since the SRATs were suppressed due to overlapping transcription in the sense direction, we asked whether these ncRNAs had overlapping transcripts (coding or non-coding). Interestingly we found that 60% of these transcripts had an overlapping annotated RNA on the opposite strand (marked as yellow; Fig. 7a).

Since SRATs were suppressed by the addition of H3K36 methylation over their start sites, we postulated that a similar addition of the methyl mark over these ncRNA promoters results in their suppression. To prove this hypothesis we used H3K36 trimethylation ChIP-Seq data from a recent publication[31]. We chose this data set over our ChIP-microarray data for its depth of coverage over promoter regions. We plotted the distribution of H3K36 trimethylation over transcribed regions and 500 bp upstream and downstream of selected classes of ncRNA transcripts compared with that over a similar region around SRATs. The H3K36 trimethylation profile shows elevated levels over the promoters of the subset of XUTs that show a significant change in expression in a *set2* (Fig. 7b, blue trace). While the levels were lower than what was seen over the promoters of SRAT genes (Fig. 7b, green trace), they were significantly higher than the levels around all the XUTs put together (Fig. 7b, orange trace). Interestingly, the other classes of ncRNA also show a similar trend in the H3 K36 trimethylation levels over their promoters (Fig. 7c,d). Therefore, we conclude that transcription of overlapping sense RNA over these ncRNA promoters adds the H3K36 methylation marks, leading to their suppression.

**Coding genes regulated by interleaved transcription units**. To understand the mechanism behind the upregulation of sense gene expression in the *set2* mutant, we focused on the 92 protein-coding genes (Supplementary Data 2, Supplementary Fig. 7) that were upregulated upon loss of Set2. Interestingly, all these genes showed the presence of overlapping transcription

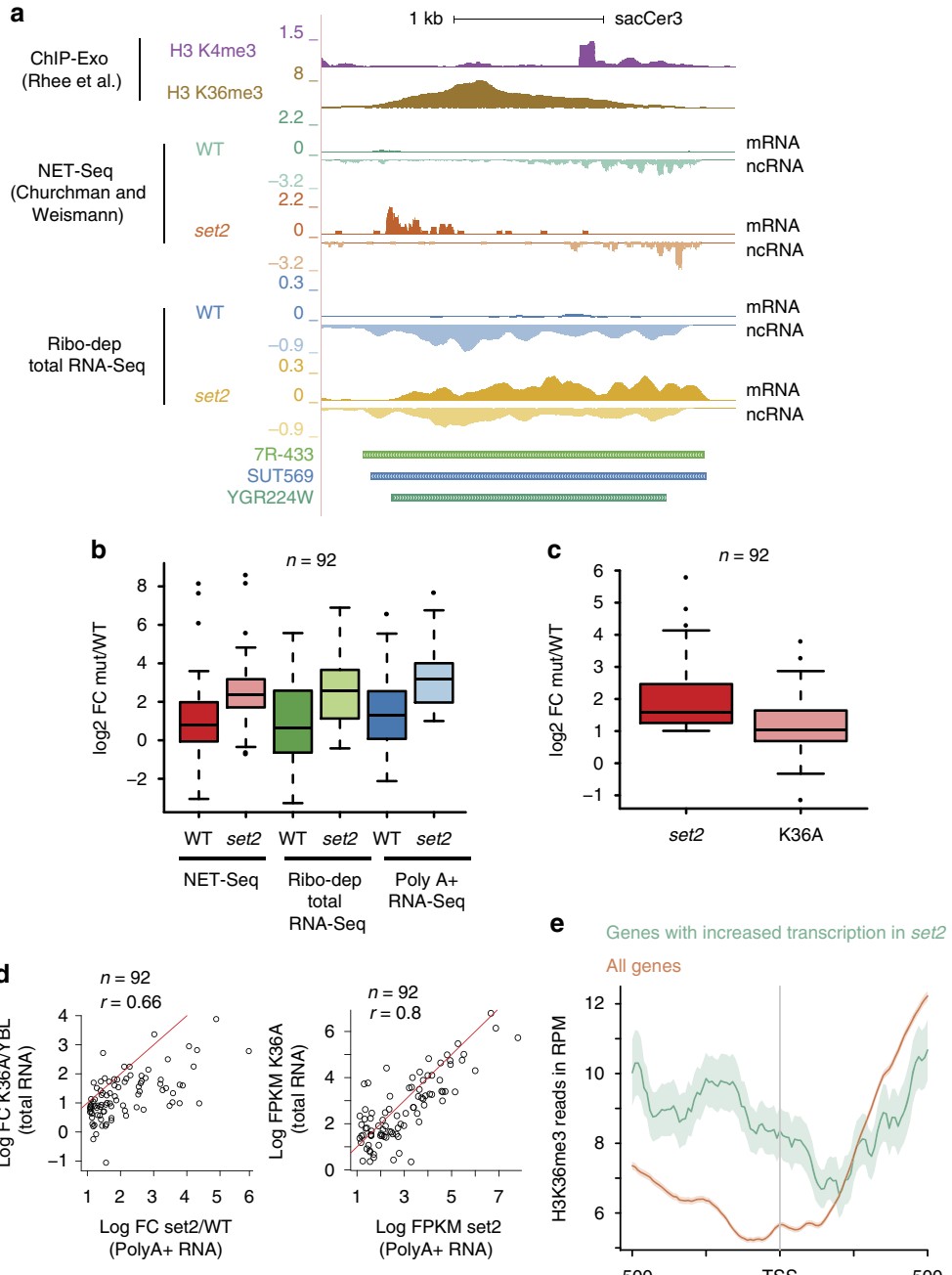

**Figure 8 | Interleaved transcription suppresses gene expression by adding H3 K36 methylation.** (**a**) Genome browser profile showing the histone modifications and transcripts (NET-Seq and ribodepleted) produced in wild-type strain (WT) and the *SET2* (*set2*) over the gene *AZR1*. The modifications are reanalysed tracks from ref. 33. Each transcript track is separated into the sense strand (mRNA) on top, running from left to right and the antisense strand (ncRNA) in the bottom running from right to left. Both strain produce a SUT or XUT that encompasses the *AZR1* gene. Loss of Set2-mediated H3K36me3 results in a de-repression of the protein-coding transcript. (**b**) Boxplot showing the abundance (RNA-Seq and PolyA-enriched RNA-Seq) or transcription levels (NET-Seq) in sense gene expression (log2 FC mut/WT) of 92 genes with overlapping transcripts in the *SET2* deletion (*set2*) versus the wild-type for each data set as indicated. (**c**) Boxplot showing the fold-change in sense gene expression (log2 FC mut/WT) of 92 genes with overlapping transcripts in the *SET2* deletion (*set2*) and H3K36A (K36A) mutants for each data set as indicated. (**d**) Scatter plots either denoting the fold change (Log FC) in gene expression (left) or the transcript abundance (Log FPKM, right) of the 92 protein-coding genes with overlapping transcripts. The respective values from the *SET2* deletion mutant are distributed on the x axis, while those from the H3K36A mutant are distributed on the y axis. The red line denotes the values where x = y. The Spearman coefficient of correlation is provided. (**e**) Metagene plots denoting the distribution of H3K36me3 mark around the promoters (upto 500 bp on either side) of all protein-coding genes (orange) or a subset of genes that are upregulated upon loss of Set2 (green). The lighter areas surrounding the traces denote the 95% confidence interval of the traces.

in the form of sense–antisense pairs or tandemly arranged genes, with the significantly altered gene arranged downstream in the pair. *AZR1* (Fig. 8a, mRNA) is an example of a sense–antisense interleaved pair of genes, where the expression of a SUT (SUT 569; Fig. 8a, ncRNA) in the wild-type strain suppresses the expression of the gene (Fig. 8a, ribodepleted total RNA Seq tracks, *set2* versus wild-type). Interestingly, the expression of the SUT is confirmed by the K4–K36 methylation distribution[32], obtained from data published from the Pugh Lab[33]. This clearly shows H3K36 trimethylation over the promoter of the *AZR1* gene. Interestingly, loss of functional Set2 in the deletion mutant, results in the de-repression of the *AZR1* gene, despite the presence of the SUT (Fig. 8a). All 92 genes in this group show a transcriptional de-repression upon *SET2* deletion (Fig. 8b), including an increase in the NET-Seq signal. The H3K36A mutant also shows a similar, albeit muted increase in the expression of these 92 genes (Fig. 8c). As mentioned previously, this observation is due to the leaky expression in the wild-type histone shuffle strain. Consistent with this explanation, RNA abundance of these 92 genes correlated strongly (Pearson's coefficient of correlation = 0.8) between the two mutant strains (Fig. 8d, right), while the fold-change in the mutants was biased towards the *SET2* mutant (Fig. 8d, left). We analysed the distribution of H3K36 trimethylation over the promoter in these 92 genes versus all the protein-coding genes in the wild-type strain. Indeed, high levels of H3K36 trimethylation over the promoters of these genes (Fig. 8e), confirms the repressive effect of the mark.

The arrangement of sense–antisense interleaved units results in suppression of antisense transcripts (SRATs) by the co-transcriptional addition of the H3K36 methyl mark. Similarly, ncRNA transcription over the promoter of a protein-coding genes also suppresses its expression through a similar mechanism. To validate the mechanism of transcription-mediated regulation of overlapping transcripts, we used the SUT569-*AZR1* sense–antisense pair of overlapping transcripts. As expected, the *set2* mutation results in *AZR1* transcription despite the presence of the SUT569 transcript (Supplementary Fig. 7B). We deleted the SUT569 promoter, thereby preventing transcription over the *AZR1* promoter, resulting in the upregulation of the *AZR1* transcription, in the presence of Set2 methyltransferase (Supplementary Fig. 7B). This establishes co-transcriptional H3K36 methylation over the *AZR1* promoter as a mechanism of transcriptional repression.

## Discussion

Set2-mediated H3K36 methylation is a repressive mark added over gene bodies during transcription elongation. However, it is a selective mark, suppressing transcription initiation from promoters within overlapping transcription units, without affecting ongoing transcription elongation. Artificially tethering Set2 methyltransferase enzyme to an activator and directing its localization to gene promoters has been shown to result in transcriptional repression[26]. The H3K36 methylation mark suppresses the expression of a novel class of antisense transcripts that we have termed SRATs. These SRATs are generated from chromatin regions that are enriched for the H3K36 methyl mark, predominantly the 3′ end of longer genes. In addition to suppressing the SRATs and other antisense ncRNA, this mechanism is employed to suppress overlapping tandemly arranged genes, where the 3′ end of the upstream gene overlaps the promoter of the downstream gene. It is interesting to note that transcription on one strand adds the mark on histones to prevent initiation from the other strand. Thus, strand-independent addition of histone marks on nucleosomes

may result in different outcomes on transcription from each strand.

Organizing the yeast genome into overlapping and interleaved transcription units utilizes the process of RNAPII transcription to recruit the Set2 methyltransferase to add the H3K36 methyl mark and regulate gene expression. An interesting consequence of linking H3K36 methylation-dependent transcriptional suppression on one strand to transcription on the other is the removal of regulatory burden from the synthesized RNA. Interestingly, the SRATs are targeted for immediate degradation. Loss of the nuclear Rrp6 or nuclear Xrn1 RNA-degradation pathways in a *SET2* deletion mutant enhances the stability of SRATs.

Our observation that a subset of CUTs, SUTs and XUTs are suppressed by H3K36 methylation suggests that ncRNA production is regulated both at the level of transcription—by chromatin regulatory factors—and RNA stability—determined by the access of RNA to the various RNA-degradation machineries. While our observation that a significant number of SRATs are polyadenylated would suggest longer half-lives and possible cytoplasmic localization, most are targeted for degradation. The function of the polyadenylated SRATs in the cytoplasm is currently unknown, but they may be subjected to degradation by the nonsense-mediated decay pathway, as is the case with some CUTs and SUTs[34]. While our analysis focused on the role of non-coding antisense transcription in regulating transcription initiation *in cis*, we cannot speculate on the possible effect of the ncRNA *in trans* based on our data.

Our analysis has identified a number of novel transcripts that are suppressed by Set2-mediated H3K36 methylation. The annotation of these novel transcripts is a crucial first step in determining their function in the regulation of various cellular processes in yeast. Since these novel non-coding transcripts have the potential to be a part of gene regulatory circuits[35,36], we carried out GO term analysis of genes with SRATs buried in their coding regions. Interestingly, we found the enrichment of a number of stress response genes, particularly in the osmoregulatory pathway and DNA-damage response pathways (Supplementary Fig. 8). The exact nature and role of antisense transcripts (or transcription) in the regulation of these genes is currently unclear.

H3K36 methylation has been shown to be involved in a number of nuclear processes including DNA damage response[37–40], RNA splicing[41], organismal development[27] and aging[42]. A number of these processes are also controlled by ncRNAs[43], working on multiple levels to regulate cellular functions. Interestingly, we found about 30% of genes identified in an aging study[42] produce SRATs upon the loss of Set2 (Supplementary Data 3). We also analysed previously published strand-specific RNA-Seq data comparing old versus young cells[44]. We found that 210 SRATs were upregulated in the old cell in comparison with the young cells (Supplementary Data 4). Further work on this subject could help clarify the molecular mechanisms involving epigenetic marks, chromatin structure and RNAs to achieve cellular function. A number of antisense transcripts respond to environmental stimuli[20,45] and have been implicated in the development of diseases. While current studies focus on the how these aberrant RNAs function in diseases, less attention is given towards understanding the mechanism behind their altered expression. Here we describe the role of Set2-mediated H3K36 methylation in suppressing ncRNA. Given the involvement of Set2 in carcinogenesis[46] and tumour heterogeneity[47,48], a deeper understanding of its molecular function is imperative. Finally, our study indicates a crucial role of genomic organization of genes in regulating transcription. Overlapping and interleaved gene organization may serve as precursors for regulatory gene circuits that use co-transcriptional marks for regulation.

## Methods

**Yeast strains.** All *S. cerevisiae* strains used in this study are listed in Supplementary Data 5. Yeast strains were grown in the appropriate selection plates. Cells were grown overnight in YPD at 30 °C, diluted to an OD600 of 0.2 in a total volume of 200 ml and collected when the OD600 reached 0.8. Cells were harvested onto a 0.45 μm nitrocellulose filter using a filtration apparatus, washed with ice cold 1× PBS and stored in −80 °C till total RNA preparation.

**Preparation of total RNA from yeast.** Total RNA was prepared using the hot-phenol method as described previously[13]. Briefly, cells were resuspended in AE buffer (50 mM sodium acetate pH 5.2 and 10 mM EDTA). SDS was added to a final concentration of 1.6% (w/v). An equal volume of water-saturated phenol was added and incubated at 65 °C for 10 min with intermittent vortexing. The samples were cooled on ice before the debris was removed by centrifugation at 13,000 g. The supernatant was poured into a 50 ml Phase Lock gel tube, mixed with an equal volume of chloroform and spun at 3,000 r.p.m. for 10 min. The aqueous supernatant containing the RNA was precipitated using isopropanol and sodium acetate. The RNA was washed with 70% ethanol and dissolved in an appropriate amount of 10 mM Tris pH8.0 prepared in DEPC water. The RNA was quantitated on a NanoDrop machine and its purity estimated using the absorbance ratio at 260 nm over 280 nm.

For polyA selection, 4 μg of total RNA in 750 μl of RNAse-free water was incubated at 65 °C for 10 min and added to 75 mg of oligo dT cellulose (Ambion, #AM10020), pre-washed and resuspended in an equal volume of 1× NETS buffer (625 mM NaCl, 10 mM Tris.HCl, pH 7.4, 10 mM EDTA, and 0.2% SDS), for 1 h on a rotator at room temperature. This mixture was poured into a disposable Bio-Rad column and allowed to settle. The mixture was gently washed five times with 1× NETS buffer. The mRNA was eluted twice with 1× ETS buffer (10 mM Tris.HCl, pH 7.4, 10 mM EDTA, and 0.2% SDS) heated to 65 °C. The eluates were pooled and the RNA was precipitated by standard techniques, dissolved and quantitated.

**Strand-specific RNA-Seq.** Total RNA isolated from the specified strains was ribodepleted using the Ribo-Zero™ Magnetic Gold Kit (Yeast) (Epicenter, cat# MRZY1324) according to manufacturer's instructions. The depleted RNA was then used for strand-specific library construction utilizing the TruSeq Stranded Total RNA LT (Illumina, RS-122–2302). The libraries from eight samples were pooled together and run on a single lane of HiSeq2500 machine for sequencing 50 bp single end reads. Poly A+ libraries were generated using the TruSeq Stranded mRNA Kit (Illumina, RS-122–2102).

**Genome coverage.** Since identification of low-abundance novel transcripts leads to discontinuity, fragmented annotation and ambiguous end definition[49], we chose to sequence the genome to obtain a 50× coverage for all samples in this manuscript. This would indicate that each base in the genome was sequenced at least 50 times based of v3 chemistries using a HiSeq2500 sequencer (Illumina) with 50 bp single reads.

**Identification of novel transcripts.** We aligned the unique reads obtained from the *SET2* deletion strain to the yeast genome (based on sequence dated April 2011 in the Saccharomyces Genome Database (http://www.yeastgenome.org/) and was obtained from the site ftp://ftp.ncbi.nlm.nih.gov/genbank/genomes/Eukaryotes/fungi/Saccharomyces_cerevisiae/SacCer_Apr2011) and identified all transcripts arising from each strand using the Cufflinks program. A five-nucleotide read gap was used to separate different transcripts. We used the BEDtools bioinformatics suite to refine this list of novel transcripts to remove known coding transcripts and their untranslated regions[29], and previously identified pervasive transcripts like the cryptic unstable transcripts (CUTs)[4,5], stable unannotated transcripts (SUTs)[4], and the Xrn1-sensitive unstable transcripts (XUTs)[6]. A discrepancy in these annotations arises from the fact that each transcript class was identified independent of one another, resulting in duplications that have not been resolved. We avoided this pitfall by removing all previously identified transcripts in our novel RNA identification pipeline. The resultant list of 1,179 novel transcripts (Supplementary Data 6) consisted entirely of transcripts from the antisense strand of protein-coding genes. This list was further pared down by estimating the differential expression of these transcripts (using limma in R) and requiring that the following parameters were met. First, to make sure that valid and complete transcripts were being designated, we selected identified transcripts with detectable expression (FPKM>1) either in the wild-type or the *SET2* deletion strain. This requirement also ensured that transcripts with breaks in the reads were removed from the analysis. Second, transcripts with a FDR of <5% (adjusted P value <0.05) were selected. Finally, we instituted a cutoff at twofold increase in the abundance of the transcripts in the *SET2* deletion mutant over the wild-type, to select ones that are upregulated upon loss of Set2. These parameters helped us obtain a list of 853 transcripts that we named SRATs.

**MNase-seq and nucleosome occupancy.** Wild-type and *set2* mutant were grown in 200 ml of YPD, crosslinked and treated with zymolase to spheroplast the yeast

cells. The spheroplasts were digested with MNase as described earlier[50]. Briefly, the crosslinked cells were resuspended in Buffer Z (1 M Sorbitol, 50 mM Tris HCl pH 7.4) with 10 mM β-mercaptoethanol. Zymolase (10 mg ml−1 in buffer Z) was added to a final concentration of 0.5 mg per ml and incubated at 30 °C with continuous mixing in a water bath. After >90% of the cells have spheroplasted, the mixture was pelleted at 7,000 g, and resuspended in 2 ml of NP buffer (1 M Sorbitol, 50 mM NaCl, 10 mM Tris 7.4, 5 mM MgCl2, 1 mM CaCl2) with 500 μM spermidine, 1 mM B mercaptoethanol and 0.075% NP-40. Three 600 μl aliquots of cells were used to titrate MNase (Worthington, 20 U per μl in 10 mM Tris pH 4.0) amounts and incubated at 37 °C for 20 min. to achieve the formation of 80% of mononucleosomes. The reactions were stopped with 1× STOP buffer (1% SDS and 10 mM EDTA). The samples were treated with Proteinase K (20 mg per ml) at 65 °C, removed the RNA and purified the DNA by phenol:chloroform extraction and ethanol precipitation. The purified DNA was subjected to paired-end 50-base reads using HiSeq2500 sequencer. The reads were aligned to the sacCer3 genome using bowtie2 with default settings. The paired reads were filtered (100- to 200-base fragments only) and centred. The reads were visualized using ngsplot. Normalized coverage genome browser tracks were generated in R.

**Data analysis.** Data were analysed using the R package. Differential expression analysis was carried out using the limma package in R. All plots and graph were created using inbuilt codes in R.

**Annotating and naming SRATs.** The SRATs were sorted according to their chromosomal location, irrespective of the strand they are on and numbered chronologically. The full list of SRATs is provided in Supplementary Data 6.

**Analysis of NET-Seq data.** NET-Seq raw data were downloaded from public databases and aligned to the yeast genome. Reads for SRATs and protein-coding genes were extracted and used for analysis using existing codes in R.

**Identification of bi-directional SRAT promoters.** Coordinates of genomic regions 500 bp upstream of the SRAT start site on the sense strand were used to score the NET-Seq data. The data were filtered for a twofold change in NET-seq signals and a minimum log2 NET-Seq signal of 0.5 RPKM in the *set2* mutant.

**Strand-specific northern blots.** For validating the production and confirming SRAT expression, we carried out strand-specific northern blot analysis using riboprobes. We selected a few SRATs based on a fold change in expression greater than twofold in a *SET2* mutant compared with the wild-type and an SRAT length >1,500 nt. These parameters were chosen to ensure that these low-abundance transcripts could be reproducibly detected in the northern blots. Multiple PCR primers were generated for each SRAT, such that the length of each probe ranged from 500–800 bp. Depending on the strand targeted by the riboprobe, a T7 promoter primer was added to the ends of the either the forward or reverse primer. The probe was amplified using yeast genomic DNA using the Taq DNA polymerase. The amplicons were purified and estimated. 1 μg of the probe DNA used to prepare the riboprobes using the MAXIscript Kit (Thermo Fisher, #AM1312M) with 32P labelled UTP (Perkin Elmer). The labelled probed were purified by using microspin G50 columns to remove unincorporated radionucleotides. The primers used to prepare the riboprobes are listed in Supplementary Data 7.

Northern blots were carried out with either 20 μg of total RNA or 2 μg of polyadenylated RNA, as described previously[13]. Briefly, the RNA mixed with ethidium bromide (10 mg ml−1) was separated on a 1.5% agarose gel running in 1× MOPS buffer (20 mM MOPS pH 7.0, 2 mM Sodium acetate and 1 mM EDTA in DEPC-treated water) with 6.6% formaldehyde. After separating the RNA, the gel was visualized with ultraviolet light and washed thrice with distilled water remove formaldehyde. The gel was then soaked in 10× SSC (1.5 M NaCl and 150 mM sodium citrate) and the RNA transferred to a Zeta probe membrane overnight. The blots were incubated at 50 °C for 2–4 h in prehybridization buffer (5× SSC, 1% SDS, 0.1% sodium pyrophosphate, 5× Denhardt's, 200 μg ml−1 sheared salmon sperm DNA, 50% deionised formamide). The riboprobes were added to the buffer after heating at 95 °C for 2 min and incubated overnight at 50 °C. The blots were washed twice at 50 °C using the wash buffer (1× SSC, 1% SDS, 0.1% sodium pyrophosphate, 50% deionised formamide) and exposed on phosphorimager cassettes from GE. The images were scanned on a Typhoon phosphorimager and quantified using the IQ software. The abundance of the SRAT was normalized to either *ACT1* or *SCR1* transcripts. Original uncropped blots of all northern blots used for the figures and the quantitation are shown in Supplementary Fig. 9.

**WCE preparation and western blot antibodies.** Whole-cell extracts for yeast were prepared according to protocol published elsewhere[25]. Briefly, 5 ml yeast cultures grown to the mid-log phase were collected, washed once with water and resuspended in 400 μl of 2 M NaOH with 8% β-mercaptoethanol. The suspension was incubated on ice for 5 min, followed by a 10 min spin at 16,000 g. The supernatant was removed and the pellet was resuspended in 400 μl of Buffer A

(40 mM HEPES-KOH pH 7.5, 350 mM NaCl. 0.1% Tween-20, 10% glycerol and protease inhibitors) incubated in ice for 10 min and harvested as above. The resultant pellet was resuspended in 2 × SDS loading dye and separated on a 15% SDS–PAGE gel. The gel was transferred to an activated PVDF membrane using the wet-transfer method. Antibodies used for chromatin immunoprecipitation (ChIP) and western blotting are H3 (Abcam, #1791), H3K36me3 (Abcam, #9050) H3K36me2 (Millipore, #07–369) and H3K36me1 (Abcam, #9048). All the antibodies were used at a dilution of 1:1,000 for the western blot. Original uncropped blots of all western blots used for the figures are shown in Supplementary Fig. 9.

**Histone shuffle to generate the H3K36A mutant.** The FY2162 strain[51] was renamed the YBL strain, in which a single copy of *HHF2-HHT2* is provided on plasmid covering the deletions of both genomic copies of the histone genes. An auxotrophic marker switch was used to replace the wild-type copy of histones with the H3K36A mutant. The H3K36A-pDM18 plasmid was obtained from the Stowers Molecular Biology Facility as a part of the Shima mutant library[52]. The wild-type or the mutant plasmids were transformed into the FY2162 yeast strain and selected on -trp-ura SD plates. Following two rounds of selection on these plates, the clones were subsequently grown on -trp + 5-FOA SD plates for two additional rounds to remove the Ura3 containing wild-type plasmid. The strain was then grown in YPD to prepare RNA used in RNA-Seq assays.

**ChIP data analysis.** H3 K36 trimethylation ChIP was analysed from our previous publication[13] and a recent paper from the Steinmetz lab[31]. The microarray data were analysed using R codes while the Seq data were analysed using ngs.plot[53].

**Data availability.** The Sequencing data generated during this study are available at NCBI SRA Browser (http://trace.ncbi.nlm.nih.gov/Traces/study/?acc=SRP089706 &go=go) under accession number SRP089706. The NET-Seq data were downloaded from GEO database with the accession number GSE 25107. The H3K36 trimethylation ChIP-Seq data were downloaded from ENA database with the accession number ERP007035. Original data underlying this manuscript can be accessed from the Stowers Original Data Repository at http://www.stowers.org/ research/publications/libpb-1115. All other original data can be obtained from the authors upon reasonable request.

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

## Acknowledgements

We thank Steve Buratowski and Tae-Soo Kim for helpful discussion. We thank Fred Winston for the histone shuffle strain, plasmids and for his input on the manuscript, and Scott Briggs for the Set2 yeast expression plasmid. We also thank Anoja Perera, Allison Peak, Kate Malanowski, Michael Peterson and the Stowers Molecular Biology Facility for sequencing the samples. This work was supported by NIH grants RO1GM047867 and R35GM118068 and support from the Stowers Institute for Medical Research.

## Author contributions

S.V. and J.L.W. conceived this study. S.V. designed and carried out all the experiments in this study. H.L. and M.M.G. wrote scripts used for the bioinformatics analysis. S.V. and H.L. carried out the data analysis. S.V. and J.L.W. wrote the manuscript.

## Additional information

**Competing financial interests:** The authors declare no competing financial interests.

