## [Peer Review File · Nature Communications]

Reviewers' comments:

Reviewer #1 (Remarks to the Author):

Previous reports by Workman and others have demonstrated that an important function of chromatin in general, and H3K36 methylation in particular, is the prevention of spurious, noncoding transcription. In this study, Venkatesh et al. expand upon their earlier work on the Set2-dependent chromatin re-setting pathway and show that deletion of Set2 gives rise to abundant, antisense transcription. These antisense transcripts occur within coding regions and appear to initiate at sites that normally are enriched for H3K36 methylation. In most cases, the occurrence of these noncoding RNAs, SRATs, in the set2 mutant background does not correlate with an effect on sense RNA synthesis. However, for a subset of genes where transcription units overlap, loss of Set2 upregulates sense transcription, in line with models wherein H3 K36 methylation is established in promoter regions of protein-coding genes by antisense transcription and loss of this modification opens chromatin and activates sense transcription.

This is a well-executed study, with thoroughly analyzed RNA seq datasets from set2 and H3K36A mutants. Follow up strand-specific northern support the RNA seq data and are of high quality. The main concerns are lack of novelty and the overall descriptive nature of the work. The Workman group and others have convincingly shown in other papers that Set2 represses cryptic initiation within coding regions. The current study is mainly an expansion of earlier work with an emphasis on antisense transcription that arises in the absence of the Set2 pathway. In addition, Churchman and Weissman in their original NET-seq experiments found extensive antisense transcription in a set2 deletion strain. Whether these antisense transcripts should be viewed as a new type of noncoding RNA is debatable. Other major classes of noncoding RNAs (CUTs, SUTs, XUTs, NUTs) are generally enriched when RNA-processing/degradation pathways are eliminated. This implies that they are expressed in wild type cells but are undetectable because of active degradation pathways. The SRATs found here are only produced in the context of the set2 or H3K36A mutant.

Specific points:

1. The authors used strains that retain RNA processing factors, such as the exosome subunit Rrp6, that rid cells of many noncoding transcripts. It's likely they are underestimating the amount and effects of antisense transcription in the set2 mutant by not enriching for these and other noncoding RNAs. This is a weakness of the study.
2. The complementation experiment (Figure 2) is basically a control that should be moved to the supplement.
3. Line 238: It is incorrect to use the term dominant when describing the results of Figure 2. The cells are haploid for all genes.
4. Supplementary Figure 2C: Why is there a faint signal for H3 K36 methylation in the set2-pRS strain?
5. Abstract: The word "unfettered" implies out of control or rampant antisense transcription. This doesn't seem to explain the results. Also, by many definitions, Set2 wouldn't be characterized as an elongation factor.
6. What does "PPN1" stand for in the figures?

7. Figure 4C: the shading around the lines needs to be defined in the legend.
8. Line 195: It's not particularly "interesting" that large amounts of rRNA cause other RNAs to migrate aberrantly in gels. This is commonly observed.
9. Line 213: "stains" should be changed to "strains"
10. Mutant genes should be written in lower case in the text and the figures.
11. Figure legend 8 needs to be corrected to match the panel labeling in the figure.
12. Line 580: The number 10 needs units.

Reviewer #2 (Remarks to the Author):

The paper by Venkatesh et al. aims at describing a novel class of lncRNAs that can be revealed in the absence of the H3K36 histone mark deposited by Set2p. After the extensive description of new transcripts, the authors shortly give some clues about their implication in global gene regulation and propose a model in which strand-independent H3K36 methylation results in different transcriptional outcomes from each strand.

First, the authors show that in a SET2 deletion strain, a class of 853 transcripts (according to their threshold) is derepressed. Surprisingly, they are all antisense to genes (and not bi-directional or only sense, see below). They call them SRATs (Set2-Repressed Antisense Transcripts). Most of them are polyadenylated. Then, they show that the derepression phenotype can be rescued by inserting SET2 on a plasmid. Although interesting, this result may be presented as supplementary data. Importantly, they show that mutation of H3K36 to alanine gives rise to the SRATs just as deletion of SET2 confirming that it is the modification itself that is essential for the repression of these ncRNAs. However, these results could be incorporated into Figure 1 to avoid the redundancy that appears in the first 3 Figures.

They then describe the SRATs with respect to the (main) different classes of lncRNAs known in *S. cerevisiae*, i.e. the CUTs, SUTs and XUTs. Here we learn that SRATs most of the time do not reach the promoter of their sense genes, possibly explaining the lack of effect on global gene expression. They mention in the text that in some cases, SRAT transcription extends into the promoter. Do these events correlate with an effect on sense expression? Those "antisense genes" (Figure 4C) (that should be named "SRATs-containing genes" to avoid confusion with antisense CUTs, SUTs or NUTs) are enriched in H3K36 methylation suggesting that the presence of this mark represses spurious transcription initiation.

Workman and colleagues then use the data of nascent transcription (NET-Seq) published by Churchman et al. NET-Seq gives more accurate measure of transcription than RNAseq. Importantly, NET-Seq analyses at the PPN1 gene in WT vs Dset2 (Figure 5A) not only show increased antisense transcription but also an increase of the sense signal at the 3' of the gene. This suggests that a bidirectional promoter is created in the gene body producing a stable SRAT and an unstable divergent RNA that is rapidly degraded. Is this a general feature? And is a nucleosome depleted region created in the gene body when SET2 is deleted? This is an important question because it challenges the idea of the authors that a strand-independent modification has a strand-specific outcome on transcription initiation. Along these lines, it is not unexpected that loss of Set2 primarily increases intragenic non-coding transcription and more rarely coding transcription, since H3K36me3 mostly peaks over coding regions and not at gene promoters.

They then show that global gene expression is not affected although the expression of some SUTs and XUTs may be increased in the SET2 deletion mutant.

At last, they focus on 92 genes that are derepressed when SET2 is absent. The authors claim that these genes present antisense or upstream tandem transcription. Although they show that H3K36me3 is increased at the promoter of these genes compared to all genes, a figure with a metagene RNAseq analysis would strengthen the idea that repression depends on H3K36me3 deposited by non-coding transcription.

This manuscript is well written and the experiments and results are clearly described but the take-home message remains very descriptive and the importance of Set2 and H3K36me3 in repressing intragenic cryptic transcription is not really novel (Carrozza et al., 2005; Venkatesh et al., 2012). Moreover, the proposition that a the strand independent addition of H3K36 methylation results in a strand-specific difference in transcriptional outcomes may be overstated. This work would therefore be better suited for a more specialized journal. Compared to NUTs, CUTs and XUTs, SRATs appear to be more a consequence of the Dset2 mutation rather than lncRNAs involved in gene regulation (although NUTs, CUTs and XUTs only affect the expression of a small set of genes). One thing that may improve the message would be to find natural conditions in which SRATs accumulate, in order to strengthen the idea that what is observed is not artificial. For example, do they appear in the aging study of the Tyler lab (Hu et al., 2014) or in the study of the Berger lab (Sen et al., 2015)?

Minor comments:

Figure 1A legend: explain the meaning of the color code for YDR452W and PPN1.

p.17: " We then selected the genes with significant change in expression (Fig. 6C) (marked as red)". Nothing is marked red in Fig. 6C.

STOWERS INSTITUTE[®]
FOR MEDICAL RESEARCH

Response to the Reviewer's Comments

Venkatesh et al., Manuscript NCOMMS-16-09176

We thank the reviewers and editors for giving us the opportunity to revise our manuscript. All the points raised by the reviewers were valid, and in addressing these points we have uncovered interesting observations that we have included in the revised manuscript. We would like to thank the reviewers for raising these points that have helped us to enhance our story further. In order to ease the review process, we have highlighted the changed text in the revised manuscript. We hope that you will find the revised manuscript acceptable for publication.

Reviewers' comments:

Reviewer #1 (Remarks to the Author):

We thank the reviewer for their comments and suggestions. In addressing all the questions raised, we hope we have improved the quality of the manuscript, and that the reviewer would find the revised manuscript acceptable for publication. The point-by-point answers to all the issues raised by the reviewer is enumerated below.

Previous reports by Workman and others have demonstrated that an important function of chromatin in general, and H3K36 methylation in particular, is the prevention of spurious, noncoding transcription. In this study, Venkatesh et al. expand upon their earlier work on the Set2-dependent chromatin re-setting pathway and show that deletion of Set2 gives rise to abundant, antisense transcription. These antisense transcripts occur within coding regions and appear to initiate at sites that normally are enriched for H3K36 methylation. In most cases, the occurrence of these noncoding RNAs, SRATs, in the set2 mutant background does not

correlate with an effect on sense RNA synthesis. However, for a subset of genes where transcription units overlap, loss of Set2 upregulates sense transcription, in line with models wherein H3 K36 methylation is established in promoter regions of protein-coding genes by antisense transcription and loss of this modification opens chromatin and activates sense transcription.

This is a well-executed study, with thoroughly analyzed RNA seq datasets from set2 and H3K36A mutants. Follow up strand-specific northern support the RNA seq data and are of high quality. The main concerns are lack of novelty and the overall descriptive nature of the work. The Workman group and others have convincingly shown in other papers that Set2 represses cryptic initiation within coding regions. The current study is mainly an expansion of earlier work with an emphasis on antisense transcription that arises in the absence of the Set2 pathway. In addition, Churchman and Weissman in their original NET-seq experiments found extensive antisense transcription in a set2 deletion strain.

We thank the reviewer for their succinct summary of our work and its appraisal. While the phenomenon of cryptic transcript generation is an established one, the identity and genomic location of transcripts that are generated was unknown. We employed the next generation sequencing methods to identify these RNA species (strand specific RNA sequencing). Therefore, this study provides the genomic co-ordinates of antisense RNA generated upon loss of H3 K36 methylation, in addition to the molecular mechanism of overlapping transcription that we have defined here. This forms a valuable resource for future studies, which can then employ these co-ordinates to determine the expression of the SRATs in specific studies.

Whether these antisense transcripts should be viewed as a new type of noncoding RNA is debatable. Other major classes of noncoding RNAs (CUTs, SUTs, XUTs, NUTs) are generally enriched when RNA-processing/degradation pathways are eliminated. This implies that they are expressed in wild type cells but are undetectable because of active degradation pathways. The SRATs found here are only produced in the context of the set2 or H3K36A mutant.

We thank the reviewer for raising a very important point. One of the features of SRAT expression that we have established in this study is its dependence of the removal of the H3 K36 methyl mark. The way we chose to show this was by using Set2 deletion mutants and the H3K36 A mutant, where the levels of H3K36 methylation are non-existent. While these mutants do not exist in nature, H3K36 methylation is dynamic in the wild-type cell, with specific de-methylases that act to reduce the various methylation levels. Thus, it is conceivable that the action of these demethylases in specific stress conditions, may lead to the removal of H3K36 methylation over the start sites of the SRATs leading to their production without affecting Set2. Interestingly, a number of publications have reported the loss of H3 K36 methylation upon aging. We analyzed two studies that reported strand specific RNA Seq data – and found a number of SRATs that are upregulated in the aged cell compared to the young yeast cells (Supplementary Tables 5 and 6). This observation underscores the importance of H3 K36 methylation in controlling the process of aging. However, further work needs to be done in order to decipher the exact role played by the ncRNAs or its transcription on the process of aging.

In addition to this, a number of mutations have been identified in the human counterpart of Set2, that affect its catalytic activity and result in a diseased state. Thus, the identification and characterization of these antisense RNA may lead to the establishment of a novel paradigm of ncRNA function in disease development.

Specific points:

1. The authors used strains that retain RNA processing factors, such as the exosome subunit Rrp6, that rid cells of many noncoding transcripts. It's likely they are underestimating the amount and effects of antisense transcription in the set2 mutant by not enriching for these and other noncoding RNAs. This is a weakness of the study.

We thank the reviewer for this comment. We carried out strand specific RNA sequencing on total RNA from deletions of components of RNA degradation pathways – the nuclear Rrp6

and the cytoplasmic Xrn1, either singly or combination with a Set2 deletion. We show that the loss of Rrp6 or Xrn1 alone result in stabilization of most SRATs, with a few being selected as significantly over-expressed based on our cutoffs. However, this number is low compared to that obtained upon loss of Set2. Based on this result we conclude that SRATs are actively transcribed as suggested by our analysis (Figure 4D), albeit at low levels in the wild-type strain. However, the resultant RNA species are not stable and are subject to both nuclear and cytoplasmic degradation. Interestingly, loss of either component of the degradation machinery in combination with the Set2 deletion, enhances the stability of a majority of SRATs, suggesting that these RNAs are actively marked for degradation both in the wild type and the *set2* mutant. We have also validated these results using the strand specific northern blot analysis. We have included these results in our revised manuscript (Figure 5 and Supplementary Fig. 6). Thanks to the reviewer's suggestion we have managed to enhance the quality of the manuscript with regard to the point raised.

As far the identification of novel transcripts, we did not observe an increase in the number of SRATs upon deletion of either Rrp6 or Xrn1. Deleting these genes along with Set2, enhanced the stability of the SRATs, improving the statistical parameters, thereby resulting in more SRATs being selected with our stringent cutoffs. This also validates the strength of our bioinformatics pipeline for the discovery of novel transcripts.

2. The complementation experiment (Figure 2) is basically a control that should be moved to the supplement.

We thank the reviewer for their suggestion. We have now moved this figure to the supplement and is marked as Supplementary Figure 3.

3. Line 238: It is incorrect to use the term dominant when describing the results of Figure 2. The cells are haploid for all genes.

We thank the reviewer for pointing this out. We have changed 'dominant' to 'central' to define the role played by Set2 in regulating intragenic transcription.

4. Supplementary Figure 2C: Why is there a faint signal for H3 K36 methylation in the set2-pRS strain?

We thank the reviewer for picking up this fine point. We have seen that the H3K36me3 antibody used in the study (Abcam # 9050) shows a very mild cross-reactivity with H3 under the conditions used in our Western blot analysis. Interestingly, the same band shows up in the western blot using the H3K36A mutant, which is devoid of H3K36 methylation. Since the H3K36me2 and H3K36me1 antibodies do not show any band in the mutants, we can conclude that this is not the result of contaminating wild type cells.

5. Abstract: The word "unfettered" implies out of control or rampant antisense transcription. This doesn't seem to explain the results. Also, by many definitions, Set2 wouldn't be characterized as an elongation factor.

We thank the reviewer for raising this issue. We agree with the reviewer and have altered the abstract accordingly. In lieu of calling Set2 a elongation factor, we have altered it to 'chromatin resetting factor', which we have established in previous publications from the Workman lab.

6. What does "PPN1" stand for in the figures?

We thank the reviewer for pointing out this omission. It is the standard name for YDR452W –which is an endopolyphosphatase.

7. Figure 4C: the shading around the lines needs to be defined in the legend.

We thank the reviewer for pointing out this omission. We have included this in the figure legends of the revised manuscript. The shading represents the 95% confidence interval.

8. Line 195: It's not particularly "interesting" that large amounts of rRNA cause other RNAs

to migrate aberrantly in gels. This is commonly observed.

We thank the reviewer for raising this issue. The idea for that statement was to explain why we were seeing altered mobility of some SRATs on the same gels, when either total RNA or poly-enriched RNA were probed. We have removed the text from the main body and added a line in the figure legends for Figure 1 to help readers understand the results better

9. Line 213: "stains" should be changed to "strains"

We thank the reviewer for pointing out this typo. We have corrected this in the revised manuscript.

10. Mutant genes should be written in lower case in the text and the figures.

We thank the reviewer for pointing out the discrepancy in the gene annotations. We have made these changes throughout the manuscript.

11. Figure legend 8 needs to be corrected to match the panel labeling in the figure.

We thank the reviewer for pointing out the mistake in labelling the figure. We have made the changes in the figure legends.

12. Line 580: The number 10 needs units.

We thank the reviewer for pointing out this omission. We have included the unit (minutes) in the revised manuscript.

Reviewer #2 (Remarks to the Author):

The paper by Venkatesh et al. aims at describing a novel class of lncRNAs that can be revealed in the absence of the H3K36 histone mark deposited by Set2p. After the extensive

description of new transcripts, the authors shortly give some clues about their implication in global gene regulation and propose a model in which strand-independent H3K36 methylation results in different transcriptional outcomes from each strand.

We thank the reviewer for their comments and suggestions. In addressing all the questions raised, we hope we have improved the quality of the manuscript, and that the reviewer would find the revised manuscript acceptable for publication. The point-by-point answers to all the issues raised by the reviewer is enumerated below.

First, the authors show that in a SET2 deletion strain, a class of 853 transcripts (according to their threshold) is derepressed. Surprisingly, they are all antisense to genes (and not bi-directional or only sense, see below).

We thank the reviewer for this observation. In the manuscript, we do not claim that there are no bi-directional or sense cryptic transcripts. However, the bioinformatics methods that we have used in this manuscript will only be able to identify antisense transcripts with a high degree of confidence. The presence of the abundant sense transcription encoding protein coding genes sets up a huge background making it impossible to discriminate these transcripts from any sense cryptic transcript that may arise in the mutant. Given the low levels of transcript abundance of the cryptic transcript, any increase in the levels of sense transcription in the mutant versus the wild type is not supported statistically.

They call them SRATs (Set2-Repressed Antisense Transcripts). Most of them are polyadenylated. Then, they show that the derepression phenotype can be rescued by inserting SET2 on a plasmid. Although interesting, this result may be presented as supplementary data.

We thank the reviewer for their suggestion. We have now moved this figure to the supplement and is marked as Supplementary Figure 3.

Importantly, they show that mutation of H3K36 to alanine gives rise to the SRATs just as deletion of SET2 confirming that it is the modification itself that is essential for the

repression of these ncRNAs. However, these results could be incorporated into Figure 1 to avoid the redundancy that appears in the first 3 Figures.

We appreciate the reviewer's suggestion of clubbing the first three figures into one. While we have moved Fig. 2 to supplementary figure 3, we would like to respectfully insist on keeping the Figure 3 (the new Figure 2) intact instead of joining it with Fig. 1 in order to maintain the narrative. The wild type strain for this experiment is the histone shuffle strain, which is different from the BY4741 yeast strain in that it has only one copy of the histone gene (either WT or mutant). Given that the histone shuffle strain with the wild type copy of histones shows a slightly different phenotype compared to the BY4741 strain, which is enumerated in Supplementary figure 4, we would like to separate this figure from Figure 1 as the "wild type" controls are not the same.

They then describe the SRATs with respect to the (main) different classes of lncRNAs known in *S. cerevisiae*, i.e. the CUTs, SUTs and XUTs. Here we learn that SRATs most of the time do not reach the promoter of their sense genes, possibly explaining the lack of effect on global gene expression. They mention in the text that in some cases, SRAT transcription extends into the promoter. Do these events correlate with an effect on sense expression?

We thank the reviewer for raising this point. There are 290 genes that have an SRAT over the promoters. Most of these genes happen to be tandemly arranged genes, with the SRAT generated from one gene going over the promoter and in some cases over the neighboring gene as well, as enumerated in our example below (Figure 1). This phenomenon occurs only in the *set2* mutant as the SRATs are not produced in the wild type. For these genes with the SRAT transcribing over the promoter in a *set2* mutant, we have not observed any significant change in the levels of sense transcription upon loss of Set2 (Figure 2). This observation could be due to the fact that co-transcriptional addition of H3 K36 methylation by Set2 over the promoter of the sense transcript by the SRAT is necessary for suppression.

Figure 1: Example of a SRAT produced in the *set2* mutant that transcribes over the promoter of the sense gene.

Figure 2: The 290 SRATs that transcribe over the promoters of the genes they are embedded in do not affect the sense transcript levels. Scatter plot showing the distribution of the abundance of sense RNAs with a SRAT transcribing over the promoter in the wild-type versus the *set2* mutant.

Those "antisense genes" (Figure 4C) (that should be named "SRATs-containing genes" to avoid confusion with antisense CUTs, SUTs or NUTs) are enriched in H3K36 methylation suggesting that the presence of this mark represses spurious transcription initiation.

We thank the reviewer for their suggestion. We have altered the figure as suggested by the reviewer. The new Figure 3B replaces Figure 4C in the revised manuscript.

Workman and colleagues then use the data of nascent transcription (NET-Seq) published by Churchman et al. NET-Seq gives more accurate measure of transcription than RNAseq. Importantly, NET-Seq analyses at the PPN1 gene in WT vs Dset2 (Figure 5A) not only show increased antisense transcription but also an increase of the sense signal at the 3' of the gene. This suggests that a bidirectional promoter is created in the gene body producing a stable SRAT and an unstable divergent RNA that is rapidly degraded. Is this a general feature?

We thank the reviewer for raising this interesting suggestion. Since RNA abundance as measured by RNA-Seq has a strong signal for the sense transcript in both the wild type and the mutant (as discussed above), we could not use this data to answer the question. However, as pointed by the reviewer, the NET-Seq data is less noisy over the coding regions and hence can be used for discern whether the SRAT promoters have bi-directional transcription. We selected genomic co-ordinates of 500 bp upstream of SRAT promoters on the strand opposite to the SRAT (i.e the sense strand). We then measured the read counts over each of these regions for the NET-Seq signal. We then selected the regions that had a log₂ NET-Seq signal greater than 0.5 (to separate non-specific signals) and those that had a 2-fold change in the NET-Seq signals between the *set2* mutant and wild type. We selected a total of 310 SRATs with significant reads in their upstream regions that could be a consequence of bi-directional transcription. Of these 24 SRATs with no reads in the wild-type strain showed enhanced NET-Seq signals over these upstream regions, clearly marking them as possessing bidirectional promoters. Based on this data we conclude that a subset of SRAT promoters are bi-directional in nature, but that this may not be a general feature. However, we would like to

add that since we used bioinformatics based screening of the NET-Seq levels, we may be underestimating the number as we would not include SRATs with a milder increase in the levels of these upstream signals. We have included these results in the revised manuscript in Supplementary Fig. 5A and B.

And is a nucleosome depleted region created in the gene body when SET2 is deleted? This is an important question because it challenges the idea of the authors that a strand-independent modification has a strand-specific outcome on transcription initiation.

We carried out MNase-Seq analysis on wild-type and *set2* mutant strains and analyzed the formation of a nucleosome free region (NFR) upstream of SRAT start sites. Interestingly, a number of SRATs (almost 50% of our list) initiate from the 3' NFR. Although both the wild-type and the *set2* mutant both show the presence of a pronounced NFR, the mutant shows a further dip in the NFR (Supplementary Fig. 5D). SRATs that begin within the gene bodies do not show a pronounced NFR although the nucleosome occupancy decreases in the *set2* mutant. These results suggest that an NFR is not necessary to initiate transcription.

Interestingly, both classes of SRAT promoters show an increase in the active transcription mark H3 K4 methylation, suggesting that these promoters behave very similar to canonical promoters. We have included this data in the revised manuscript (Supplementary Figure 5E). Similar to the result of bi-directional transcription described above, we find that the NFR is not found associated with all SRAT promoters – denoting a high degree of diversity among these promoters.

Along these lines, it is not unexpected that loss of Set2 primarily increases intragenic non-coding transcription and more rarely coding transcription, since H3K36me3 mostly peaks over coding regions and not at gene promoters.

We thank the reviewer's for presenting this argument. However, this assumption is based on the genomic distribution of the mark and not actual RNA-Seq data. Set2 has long been viewed as an elongation factor, whose function was unclear. With our previous papers we established Set2- mediated methyl mark over the gene bodies as a signal to replace and re-

assemble nucleosomes over these regions as a part of a chromatin resetting pathway. Given the role of this mark on controlling splicing in higher eukaryotes, we were interested to know if this mark signaled for regulating elongation. If it did, loss of Set2 would result either in a drop in sense transcription abundances or in increased truncated sense transcripts. Both these possibilities were not observed in our data. Also with widespread overlapping transcription in yeast, it was possible that the co-transcriptional addition of H3K36 methylation would result in transcriptional suppression over other genomic regions. This manuscript provides experimental proof of these concepts.

They then show that global gene expression is not affected although the expression of some SUTs and XUTs may be increased in the SET2 deletion mutant. At last, they focus on 92 genes that are derepressed when SET2 is absent. The authors claim that these genes present antisense or upstream tandem transcription. Although they show that H3K36me3 is increased at the promoter of these genes compared to all genes, a figure with a metagene RNAseq analysis would strengthen the idea that repression depends on H3K36me3 deposited by non-coding transcription.

We thank the reviewer for this suggestion. We tried to include the requested addition to the figure, however the program we used, ngsplot, could not handle strand specific RNA Seq data input. In lieu of this we carried out an experiment on the *AZRI*-SUT569 sense – antisense pair to determine whether the loss of the SUT569 transcription results in derepression of *AZRI*. The loss of Set2 results in the derepression of *AZRI* in the presence of SUT569 transcription. We used strand specific northern blots to show that the loss of SUT569 transcription by the deletion of the SUT promoter results in the upregulation of the *AZRI* – despite the presence of active Set2 methyltransferase. These data put together suggests that SUT transcription adds H3K36 methylation over the *AZRI* promoter to repress its transcription.

This manuscript is well written and the experiments and results are clearly described but the take-home message remains very descriptive and the importance of Set2 and H3K36me3 in repressing intragenic cryptic transcription is not really novel (Carrozza et al., 2005;

Venkatesh et al., 2012). Moreover, the proposition that the strand independent addition of H3K36 methylation results in a strand-specific difference in transcriptional outcomes may be overstated. This work would therefore be better suited for a more specialized journal.

Compared to NUTs, CUTs and XUTs, SRATs appear to be more a consequence of the Dset2 mutation rather than lncRNAs involved in gene regulation (although NUTs, CUTs and XUTs only affect the expression of a small set of genes).

One thing that may improve the message would be to find natural conditions in which SRATs accumulate, in order to strengthen the idea that what is observed is not artificial. For example, do they appear in the aging study of the Tyler lab (Hu et al., 2014) or in the study of the Berger lab (Sen et al., 2015)?

We thank the reviewer for raising a very important point. One of the features of SRAT expression that we have established in this study is its dependence of the removal of the H3 K36 methyl mark. The way we chose to show this was by using Set2 deletion mutants and the H3K36 A mutant, where the levels of H3K36 methylation are non-existent. While these mutants do not exist in nature, H3K36 methylation is dynamic in the wild-type cell, with specific de-methylases that act to reduce the various methylation levels. Thus, it is conceivable that the action of these demethylases in specific stress conditions, may lead to the removal of H3K36 methylation over the start sites of the SRATs leading to their production without affecting Set2. Interestingly, a number of publications have reported the loss of H3 K36 methylation upon aging. We analyzed two studies that reported strand specific RNA Seq data – and found a number of SRATs that are upregulated in the aged cell compared to the young yeast cells (Supplementary Tables 5 and 6). This observation underscores the importance of H3 K36 methylation in controlling the process of aging. However, further work needs to be done in order to decipher the exact role played by the ncRNAs or its transcription on the process of aging.

Minor comments:

Figure 1A legend: explain the meaning of the color code for YDR452W and PPN1.

We have included an explanation in the figure legends stating that the bar shown in green includes the untranslated regions while that in blue denotes the protein coding region.

p.17: " We then selected the genes with significant change in expression (Fig. 6C) (marked as red)". Nothing is marked red in Fig. 6C.

We thank the reviewer for pointing out the mislabeled figure. We have corrected this and have included it in the Figure 7A.

REVIEWERS' COMMENTS:

Reviewer #1 (Remarks to the Author):

In the revised manuscript by Venkatesh et al., the authors provide additional support for the discovery of Set2-repressed antisense transcripts in yeast. The manuscript has been improved by the analysis of these transcripts in strains lacking RNA decay pathways. The authors further strengthen the study by analyzing SRAT initiation sites with respect to nucleosome occupancy and by deleting the promoter for an SRAT and testing the effect on sense transcription. In total, the manuscript has been improved in response to all the reviewer comments. A few minor issues remain that should be addressed prior to publication.

1. Line 93. Either "over" or "at" should be deleted.
2. Lines 171-174. I'm finding the wording in this section to be circular. SRATs are defined as transcripts that increase in *set2Δ* mutants. I think it's a matter of phrasing, but emphasizing that SRATs are more abundant in the *set2Δ* mutant than in wild type strains sounds like the SRATs were initially discovered in some other way.
3. Lines 326-327: More information and/or a better explanation is needed to help readers draw conclusions about bi-directionality from Supplementary Fig. 5A and 5B.
4. Line 416: Replace "to" with "on".
5. Line 417: Supplementary Fig. 5 should be replaced with Supplementary Fig. 7
6. Supplementary Figure 7B requires a loading control. Some of the effects appear subtle and the signals look rather weak.
7. Tables 1 and 2 require better formatting.

Reviewer #2 (Remarks to the Author):

In this revised version, the authors have answered most of the questions raised by the reviewers and the manuscript is clearly improved and more complete. This study very nicely and extensively analyses the global effects of loss of Set2 and H3K36me3 on transcription and RNA production, and generates an important dataset. However, it is very descriptive and confirms that this histone modification primarily represses intragenic transcription initiation, an observation that the authors already published several years ago. Moreover, it is still not fully clear why this modification is important, since its loss does not substantially affect the expression of most coding ORFs. Because of the lack of novel concepts or ideas, this study may be better suited for publication in a more specialized journal.

RESPONSE TO REVIEWERS' COMMENTS:

We thank the reviewers for their helpful suggestions and critical comments. These comments and suggestions have helped improve our manuscript. We thank the reviewers for recommending our manuscript for publication.

Reviewer #1 (Remarks to the Author):

In the revised manuscript by Venkatesh et al., the authors provide additional support for the discovery of Set2-repressed antisense transcripts in yeast. The manuscript has been improved by the analysis of these transcripts in strains lacking RNA decay pathways. The authors further strengthen the study by analyzing SRAT initiation sites with respect to nucleosome occupancy and by deleting the promoter for an SRAT and testing the effect on sense transcription. In total, the manuscript has been improved in response to all the reviewer comments. A few minor issues remain that should be addressed prior to publication.

We thank the reviewer for their positive appraisal of our revised manuscript.

1. Line 93. Either “over” or “at” should be deleted.

We thank the reviewer for pointing out this typo. We have removed ‘over’ and retained ‘at’.

2. Lines 171-174. I’m finding the wording in this section to be circular. SRATs are defined as transcripts that increase in set2 Δ mutants. I think it’s a matter of phrasing, but emphasizing that SRATs are more abundant in the set2 Δ mutant than in wild type strains sounds like the SRATs were initially discovered in some other way.

We thank the reviewer for pointing the circular argument. We agree with the reviewer and have altered the sentence accordingly.

3. Lines 326-327: More information and/or a better explanation is needed to help readers draw conclusions about bi-directionality from Supplementary Fig. 5A and 5B.

We thank the reviewer for pointing out the lack of clarity in understanding the experiments leading us to conclude that bi-directionality is found in a subset of SRAT promoters. Since we are well within our word limit we have expanded on this section to explain the results in more details.

4. Line 416: Replace “to” with “on”.

We thank the reviewer for this suggestion and have replaced ‘to’ with ‘on’.

5. Line 417: Supplementary Fig. 5 should be replaced with Supplementary Fig. 7

We thank the reviewer for pointing out this mis-labelled Supplementary figure reference. We have corrected it in this version.

6. Supplementary Figure 7B requires a loading control. Some of the effects appear subtle and the signals look rather weak.

We thank the reviewer for this point. We have added the loading controls and have quantitated the blot.

7. Tables 1 and 2 require better formatting.

We have provided both tables as excel files for better formatting.

Reviewer #2 (Remarks to the Author):

In this revised version, the authors have answered most of the questions raised by the reviewers and the manuscript is clearly improved and more complete. This study very nicely and extensively analyses the global effects of loss of Set2 and H3K36me3 on transcription and RNA production, and generates an important dataset. However, it is very descriptive and confirms that this histone modification primarily represses intragenic transcription initiation, an observation that the authors already published several years ago. Moreover, it is still not fully clear why this modification is important, since its loss does not substantially affect the expression of most coding ORFs. Because of the lack of novel concepts or ideas, this study may be better suited for publication in a more specialized journal.

We thank the reviewer for their positive appraisal of our revised manuscript and their honest evaluation. While we are glad the reviewers find the revised manuscript improved sufficiently, we respectfully disagree with their estimation that this manuscript reiterates our earlier data. We have identified and characterized a novel set of transcripts that may function in stress response or other specialized growth conditions. We have provided the genomic coordinates for these transcript, which will be a useful resource for the yeast research community. In addition, we have also emphasized the importance of transcription and not the transcript in regulating gene expression.